# Invisible Hand behind Female Reproductive Disorders: Bisphenols, Recent Evidence and Future Perspectives

**DOI:** 10.3390/toxics11121000

**Published:** 2023-12-07

**Authors:** Xiaoyun Wu, Yuchai Tian, Huizhen Zhu, Pengchong Xu, Jiyue Zhang, Yangcheng Hu, Xiaotong Ji, Ruifeng Yan, Huifeng Yue, Nan Sang

**Affiliations:** 1Research Center of Environment and Health, College of Environment and Resource, Shanxi University, Taiyuan 030006, China; 202023902017@email.sxu.edu.cn (X.W.); 202313905004@email.sxu.edu.cn (Y.T.); 202223905013@email.sxu.edu.cn (H.Z.); 202123905010@email.sxu.edu.cn (P.X.); 202323905009@email.sxu.edu.cn (J.Z.); 202323905004@email.sxu.edu.cn (Y.H.); sangnan@sxu.edu.cn (N.S.); 2Department of Environmental Health, School of Public Health, Shanxi Medical University, Taiyuan 030001, China; jixiaotong@sxmu.edu.cn

**Keywords:** bisphenols, uterus, ovary, female reproductive toxicity

## Abstract

Reproductive disorders are considered a global health problem influenced by physiological, genetic, environmental, and lifestyle factors. The increased exposure to bisphenols, a chemical used in large quantities for the production of polycarbonate plastics, has raised concerns regarding health risks in humans, particularly their endocrine-disrupting effects on female reproductive health. To provide a basis for future research on environmental interference and reproductive health, we reviewed relevant studies on the exposure patterns and levels of bisphenols in environmental matrices and humans (including susceptible populations such as pregnant women and children). In addition, we focused on *in vivo*, *in vitro*, and epidemiological studies evaluating the effects of bisphenols on the female reproductive system (the uterus, ovaries, fallopian tubes, and vagina). The results indicate that bisphenols cause structural and functional damage to the female reproductive system by interfering with hormones; activating receptors; inducing oxidative stress, DNA damage, and carcinogenesis; and triggering epigenetic changes, with the damaging effects being intergenerational. Epidemiological studies support the association between bisphenols and diseases such as cancer of the female reproductive system, reproductive dysfunction, and miscarriage, which may negatively affect the establishment and maintenance of pregnancy. Altogether, this review provides a reference for assessing the adverse effects of bisphenols on female reproductive health.

## 1. Introduction

The female reproductive system is primarily composed of the ovaries, fallopian tubes, uterus, and vagina, and is regulated by the hypothalamic–pituitary–gonadal axis. The stability of its structure and function is crucial for the smooth operation of complex processes such as the production of estrogen, the production and transport of high-quality ova, the successful establishment of pregnancy, and the maintenance of normal fetal growth and development [1]. Reproductive disorders are widely recognized as a major health concern worldwide. According to the World Health Organization (WHO), the prevalence rate of reproductive disorders in various countries is estimated to be 4–14% [2]. The decline in female reproductive ability may be influenced by various factors, including lifestyle, physical factors, environmental factors, and endocrine problems [3]. In recent years, multiple experimental and epidemiological studies have shown that one of the main causes of reproductive disorders is increased exposure to environmental pollutants [4]. In particular, endocrine-disrupting chemicals (EDCs) can mimic or block natural hormone activity in the human body [1]. Long-term exposure to EDCs may interfere with the development of the female reproductive system, cause reproductive tissue lesions or even cancer, damage fertility, and lead to premature menopause [5].

Bisphenol A (BPA), a high-yield monomer widely used in the synthesis of polycarbonate plastics and epoxy resins, has been detected in daily necessities such as canned food and beverages, food container liners, and medical devices, and regulatory agencies across the world have banned the use of BPA due to its serious adverse effects on reproduction, metabolism, development, and the nervous and immune systems [6]. In recent years, increased concentrations of BPA analogs have been detected, suggesting a shift toward the use of alternative substances in industrial settings [7]. However, the use of BPA in food and beverage containers and epoxy resins for commercial products remains high, with its annual production exceeding 6 billion pounds [8]. Among the Chinese population, the average estimated dietary intake of bisphenol S (BPS), bisphenol F (BPF), and BPA is 48.5–86.4 ng/kg body weight (bw)/day, 6.3–6.4 ng/kg bw/day, and 117.2–153.7 ng/kg bw/day, respectively [9]. Similarly, the daily intake of BPA for Turkish women was 170 ng/kg bw/day based on 24-h dietary review data [10], while in Taiwan, China, the daily intake of BPA for children was 201 ng/kg bw/day [11]. The BPA intake for the Portuguese population was between these two values, with adolescents and adults having a daily intake of 79.1 and 46.1 ng/kg bw/day, respectively [12]. In addition, the concentrations of BPA, BPF, and BPS in water are 0.0095–0.173 μg/L, 0.0024–0.282 μg/L and 0.0016–0.0598 μg/L, respectively [13]. Moreover, BPA and BPS have been detected in indoor dust at detection rates exceeding 80% [14]. Consequently, human exposure to bisphenol compounds is unavoidable.

With the gradual increase in research on BPA, it has been found that its chemical structure is similar to that of estradiol, and it possesses estrogenic and antiandrogenic biological activities [15]. It acts as a weak agonist binding to estrogen receptors (ER) α and β as well as androgen receptors, resulting in endocrine disrupting effects [16], as evidenced by Sohoni P et al.’s anti-androgen screening experiments using Saccharomyces cerevisiae [17]. These effects can be carried over to the offspring, e.g., exposure to BPA during gestation in rats causes damage to epithelial cell proliferation levels, androgen receptor expression, and prostate structure in male offspring [18]. It has been found that BPA competes with E2 for binding to ER in an *in vitro* cell proliferation assay of the human breast cancer cell line MCF-7, but their estrogenic effects are counteracted in the presence of ER antagonists [19]. All of the above studies have shown that BPA produces estrogenic and anti-androgenic bioactivities through binding to hormone receptors, affecting the normal function of the body’s own estrogenic or androgenic actions.

To date, numerous studies have investigated the mechanisms through which bisphenols cause damage to the reproductive system. The second Scientific Statement released by the Endocrine Society in 2015 focused on how bisphenols affect the reproductive system of exposed individuals and their offspring through physiological, cellular, molecular, and epigenetic mechanisms [1]. Ziv-Gal et al. reported that both human epidemiological and animal studies published from 2007 to 2016 confirmed the adverse effects of BPA on the female reproductive system and reproductive-related processes (such as the estrus cycle and implantation) [20]. Peretz et al. reported that BPA, a uterine and ovarian toxin, reduces oocyte quality and uterine receptivity, and is associated with changes in human hormone levels, sexual dysfunction, and implantation failure [21]. Administration of BPA/BPS to animals during pregnancy may lead to an increase in the proportion of dopamine+ trophoblast giant cells (GCs), while the proportion of serotonin+ GCs correspondingly decreases, which may affect the placental–brain axis of the developing fetus [22]. Exposure to BPB and BPAF leads to accelerated differentiation of the mammary glands (MGs), and in addition, non-targeted metabolomics results show significantly elevated amino acid levels within the maternal MGs, which can lead to an amino acid imbalance or hyperammonemia in newborns [23,24]. The imbalance of placental and breast functions leads to abnormal fetal development and altered expression of genes related to carcinogenesis in the adult offspring. BPB has similar or stronger adverse effects than BPA on the reproductive health of rats and zebrafish, resulting in reduced testosterone production and impaired sperm development [25].

Considering the rapid increase in the concentration of bisphenols in food and the environment, and the in-depth understanding of their adverse effects on health, this review aimed to provide a brief overview of the relationship between bisphenols and female reproductive health based on *in vivo* and *in vitro* studies and epidemiological data. Specifically, given the current pollution status and the exposure patterns of bisphenols, we included studies investigating the damaging effects of bisphenols on the morphology and function (the estrus cycle, uterine acceptance and spiral artery remodelling, follicle development, oocyte division and intrauterine implantation) of major female reproductive organs (the uterus, ovaries, fallopian tubes, and vagina) and elucidating the related mechanisms of action. In addition, we summarized the causal relationship between serum/urine levels of bisphenols and female reproductive system diseases (uterine fibroids (UF), endometriosis (EM), endometrial carcinoma (UCEC), polycystic ovary syndrome (PCOS), decreased ovarian reserve function and poor ovarian response (POR), decreased fertility, and increased risk of miscarriage), aiming to reach a consensus on the adverse effects of bisphenols on female reproductive health.

## 2. Search Strategies

We conducted a literature search in PubMed (https://www.ncbi.nlm.nih.gov/pubmed) and Google Scholar (https://scholar.google.com/), accessed on 26 October 2023. The keyword ‘bisphenol’ was used to search for articles focusing on the contamination status, sources, and exposure routes of bisphenols published between 2018 and 2023. Studies focusing on the relationship between bisphenols and female reproductive disorders published between 2010 and 2023 were searched using the following keywords: bisphenol, female, pregnancy, reproduction, uterus, ovary, vagina, and fallopian tubes. In addition, we screened review articles to refine the relevant information.

## 3. Contamination Status of Bisphenols

### 3.1. Sources of Bisphenols

BPA is an organic compound that is widely used in the production of polyvinyl chloride (PVC), polycarbonate plastics, and epoxy resins. In terms of production and usage, BPA is the most commonly used plasticizer worldwide [26]. Since BPA is mainly used in everyday products, exposure to BPA during infancy is unavoidable. For instance, the detection rates of BPA, BPS, BPF, BPAF, bisphenol M (BPM), and bisphenol TMC (BPTMC) have been estimated to be >50% in baby bottles available commercially [27]. In Brazil, bisphenol compounds ranging from 10.9 to 198.9 μg/kg in concentration have been detected in 22 types of infant formula [28]. Additionally, children can be exposed to BPA through plastic toys at childcare facilities [29]. In Israel, BPA migration has been reported to exceed the European Union standard in 14–45% of tests on children’s toys and products [30]. In Spain, 90.6% of infant and toddler socks have been reported to contain BPA at concentrations ranging from 0.70 to 3736 ng/g [31].

Adults are commonly exposed to BPA and BPS through more diverse sources, including food packaging, can coatings, beverage containers, thermal printers, and clothing [32]. Large amounts of BPS and alternative color developers (up to 214 μg/cm^2^) have been found in 140 types of packaging materials for fresh food in North America, whereas BPA (62.3%), BPS (20.5%), and other BPS derivatives have been found in 311 receipts and other thermal paper products in 14 countries across Europe, Asia, North America, and Oceania [29,33]. Testing on clothing has revealed that BPA is present in all new and used clothes (mean concentration, 88.4 ± 289 ng/g), with its concentration being higher in new clothes [34]. Consistently, wastewater from textile cleaning companies has high concentrations and detection frequencies of bisphenols [35]. BPF is used in dental sealants, oral restoration devices, and tissue substitutes, resulting in its high exposure to individuals with oral diseases [36]. With the development of electronic technology, the use of BPAF in the manufacture of electronic products and fiber optics has increased [37]. More than a dozen bisphenols are used in the production of white foam takeout containers, and the rapid growth of the food delivery industry has led to an increase in human exposure to bisphenols [38]. Bisphenol A diglycidyl ether (BADGE) is widely used in epoxy resin systems to formulate protective coatings for bridges and other steel structures [39]. BPA is used in automobile tires, bodies (10–300 ng/g), and brake fluids (0.3–5.5 g/L) [40]. In addition, bisphenol compounds have been detected in 90% of over-the-counter (OTC) medications, with their total concentration ranging from ND to 415 ng/g, representing an important exposure source for the elderly population [41].

### 3.2. Bisphenol Contamination in the Environment

The use of bisphenols in various products has led to their accumulation in the environment and human bodies. BPA and its analogs are widely present in surface water, groundwater, drinking water, seawater, and sewage (Table 1). Some studies have shown that the concentrations of BPA and BPS in water samples from San Francisco Bay in the United States of America are 0.73–35 ng/L and 1–120 ng/L, respectively [42]. The concentration of BPAF in Lake Taihu was 140 ng/L, even higher than that of BPA [43]. The total bisphenol concentration is as high as 7740 ng/L in rainfall runoff samples from cities in southern China, suggesting that the concentration of bisphenols in surface water is greatly influenced by rainfall [44]. On the Qinghai–Tibet Plateau, the concentration of BPA in the Lhasa River basin is 780 ng/L and 8700 ng/L during the dry and rainy seasons, respectively, which confirms the higher levels of bisphenols in precipitation [45]. The concentration of BPA in groundwater substantially varies worldwide, ranging from 0.09 to 2.28×10^5^ ng/L [46]. In Romania, the United Kingdom, and Poland (all European countries), the influent concentrations of bisphenols in sewage treatment plants are 1 × 10^5^ ng/L, 9140 ng/L, and 400 ng/L, respectively. However, the effluent concentration of bisphenols ranges from 62 to 100 ng/L, which is similar to that observed in New York, United States of America [47,48,49,50]. In China, the concentration of bisphenols is lower in treated wastewater than in untreated wastewater in sewage treatment plants, with the concentration of BPA being <10 ng/L in drinking water, most of which is adsorbed by sludge. The average concentrations of six bisphenols in sludge from the Nanjing drinking water project range from 47.5 to 353 ng/L [51,52]. In France, the detection rate of bisphenols in 323 samples of drinking water consumed by pregnant women has been reported to be 88% [53]. In addition to fresh water, seawater contains varying amounts of bisphenols. The average concentration of BPA in oceans is high worldwide (3.29 × 10^4^ ng/L), indicating more severe pollution in seawater near regions with human activities. In addition, many beaches contain harmful concentrations of BPA in the sand (average concentration, 4.25 μg/g dry weight (dw)) [54].

Bisphenols are not only present in aquatic environments but also distributed in soil and air. In agricultural soil samples from 29 cities in China, the concentrations of BPA, BPF, and BPS have been reported to be 0.17, 0.21, and 0.078 μg/g dw, respectively. These bisphenols can enter the human body through bioaccumulation [55]. A global analysis of indoor dust samples revealed the widespread presence of BPA, BPF, and BPS [56].

**Table 1 toxics-11-01000-t001:** Concentrations of bisphenols detected in the environment.

Sample Sources	Country	Measurement Time	BPA		BPS		BPF		BPB		BPAF		TBBPA		Reference
			Conc. (Range/Mean)	DR. (%)	Conc. (Range/Mean)	DR. (%)	Conc. (Range/Mean)	DR. (%)	Conc. (Range/Mean)	DR. (%)	Conc. (Range/Mean)	DR. (%)	Conc. (Range/Mean)	DR. (%)	
Bay water	America	2017	12 ng/L	97	8.8 ng/L	41									Shimabuku I et al., 2021 [42]
Surface water	Romania	2018–2019	74.5–135 ng/L		6.15–8.23 ng/L										Chiriac FL et al., 2020 [47]
River	United Kingdom	2015	38.1 ng/L												Petrie B et al., 2018 [48]
Lake	China	2016	26 ng/L	100	16 ng/L	100	78 ng/L	100	20 ng/L	100	110 ng/L	100			Liu Y et al., 2017 [43]
Rainfall	China	2015	1480 ng/L	100	3.72 ng/L	100	56.7 ng/L	100			1.38 ng/L	76			Huang Z et al., 2020 [44]
Drinking water	China	2017	1.6 ng/L	40	0.1 ng/L	25	0.04 ng/L	5	0.2 ng/L	10	0.4 ng/L	30			Zhang H et al., 2018 [51]
Coastal water	Malaysia	2022	59.01 ng/L	100	10.96 ng/L	1144	17.65 ng/L								Zainuddin AH et al., 2023 [57]
Fresh water	Europe	NF	29 ng/L												Staples C et al., 2018 [58]
Marine water			7 ng/L												
Oceanic sandy beaches	26 countries around the world	2014–2015	4247 μg/kg												Kwon BG et al., 2020 [54]
Soil	China	NF	0.17 μg/g dw		0.078 μg/g dw		0.21 μg/g dw								Xu Y et al., 2021 [55]
Dust	12 countries around the world	2012–2014	1000 μg/g		220 μg/g		1000 μg/g		<1 μg/g		3.1 μg/g		87 μg/g		Wang W et al., 2015 [56]

Conc. range: minimum–maximum concentration; Conc. mean: mean concentration; DR: detection rate; NF: not shown in the references; dw: dry weight.

### 3.3. Bisphenol Contamination in Humans

Progressively higher levels of bisphenol contaminants in the environment also significantly increase the risk of human exposure, with the earliest exposure occurring during fetal development (Table 2). In a study including 1213 pregnant women from the Netherlands (2016), the urinary concentrations of BPA, BPS, and BPF were 1.67, 0.35, and 0.58 ng/mL during early pregnancy, respectively, whereas those of BPA and BPS were 1.46 and 0.24 ng/mL during mid-pregnancy, respectively [59]. In another study involving 1379 pregnant Dutch women (2004), the urinary concentrations of BPA, BPS, and BPF were 4.93, 0.68, and 0.9 ng/mL during early pregnancy, respectively, whereas those of BPA and BPS were 5.83 and 0.13 ng/mL during mid-pregnancy, respectively [60]. These findings indicate a reduction in the exposure level of BPA among pregnant Dutch women from 2004 to 2016. However, exposure to BPS and BPF gradually increased during this period, which is consistent with the increase in the concentration of bisphenols in the environment. In a cohort study including 196 pregnant women from South Korea (2017–2019), the urinary concentrations of BPA, BPS, and BPF were found to be 2.1, 0.1, and 0.2 ng/mL, respectively [61]. In another cohort study including 318 mother–infant pairs from South Korea (2011–2012), the maternal urinary and serum concentrations of BPA were 2.86 and 1.56 ng/mL, respectively, whereas the fetal urinary and serum concentrations of BPA were 4.75 and 1.71 ng/mL, respectively. These results suggest that BPA can be transferred through the placental barrier [62]. In a cohort study including 1197 pregnant women from China (2011–2013), the detection rates of BPA, BPF, and BPS in prenatal urine were 94.4% (2.10 ng/mL), 77.1% (0.57 ng/mL), and 47.9% (0.4 ng/mL), respectively [63]. In a study including 500 pregnant women from Guangxi (2015), the detection rates of BPA, BPB, BPS, TBBPA, and BPF were 99.6 (4.73 ng/mL), 88.8% (0.243 ng/mL), 88.2% (0.09 ng/mL), 71.4% (0.78 ng/mL), and 67.2% (0.461 ng/mL), respectively. These results indicate an overall increase in bisphenol exposure from 2013 to 2015 [64]. In a cohort study including 2023 maternal samples collected from hospitals in China (2015–2018), the detection rates of all bisphenols were > 60%. In particular, the concentrations of various BPA analogs were similar to those observed in Guangxi in 2015, whereas the concentration of BPA was decreased by half. This finding suggests that the contamination status of bisphenols in China has not worsened and some progress has been made in the control of BPA use. In a study involving various populations, the lowest levels of BPA were found in pregnant women from South Africa and the United States of America. The urinary concentrations of BPA during late pregnancy were found to be 1.16 and 0.73 ng/mL, respectively, whereas those of other bisphenols were similar to those observed in other population cohorts [65,66]. A study from South Korea reported the presence of BPA at a concentration of 0.53 ng/g in the placenta, indicating that it can be transferred from the mother to the fetus through the placental barrier [62].

In a study including 150 infants from China, the detection rates of BPA in serum samples from male and female infants were 90.67% (1.15 ng/mL) and 89.33% (0.28 ng/mL), respectively, which were similar to those observed in urine samples from infants in a study from South Korea (0.94 ng/mL) [77]. The detection rates and concentrations of BPS in serum samples from male infants in the Chinese and South Korean studies were 5.33% (0.32 ng/mL) and 16.00% (0.28 ng/mL), respectively, which were higher than those observed in serum samples from female infants in both studies [78]. However, in a study from South Africa, the concentrations of BPA in umbilical cord blood samples from 60 male and female infants were 0.53 and 1.09 ng/mL, respectively, which were different from those observed in Chinese infants [65].

Furthermore, we assessed bisphenol exposure in children from various countries. In a study from South Korea (2016–2018), the concentrations of BPA, BPF, and BPS in urinary samples from 8-year-old children were 1.96, 0.40, and 0.16 μg/g creatinine (Cr), respectively [68]. In a study from China (2018), the concentrations of BPA, BPF, and BPS in urinary samples from 6-year-old children were 2.91, 1.53, and 0.02 μg/g Cr, respectively [79]. In a study from the Netherlands (2018), the concentrations of BPA and BPS in urinary samples from 6-year-old children were 2.5 and 0.13 nmol/L, respectively, which is substantially lower than that observed in Chinese and South Korean children [69]. Additionally, in studies from Slovenia and Egypt, the concentrations of BPA in urinary samples from children have been reported to be 1.9 and 0.79 ng/mL, respectively [70,71]. Among the limited number of studies assessing the concentration of bisphenols in blood, the serum concentrations of BPA, BPF, and BPS in Chinese children (2018–2019) have been reported to be 1.60, 0.08, and 0.04 ng/mL, respectively [72].

In a study including 1317 adolescents from China, the detection rates of BPA, BPS, and BPF were 97.8%, 88.4%, and 54.8%, respectively, whereas their median concentrations were 1.2 ng/mL, 0.3 ng/mL, and 0.2 ng/mL, respectively [73]. In a study including 423 teenagers (14–15 years old) from Belgium, BPF was the most commonly detected compound (97%), followed by BPA (86%) and BPS (83%). BPB, bisphenol Z (BPZ), and BPAF had detection rates of 57%, 37%, and 12%, respectively. The concentration of BPA was highest (median 1.05 ng/mL), whereas the median concentrations of BPF (0.14 ng/mL) and BPS (0.12 ng/mL) were substantially lower [74]. However, the concentrations of bisphenols detected in adolescents were lower than those observed in children and infants.

In adults, exposure to bisphenols is more likely due to social and occupational factors. Among 1046 adults in the National Health and Nutrition Examination Survey conducted in the United States of America (2013–2016) and 3268 adults in the Korean National Environmental Health Survey (2015–2017), the urinary detection rates and concentrations of BPA were similar at 96.8% (1.23 ng/mL) and 98.1% (1.27 ng/mL), respectively [75]. In a study including 144 adults from Norway, the detection rates of BPA, BPF, and BPS in urine samples were 96% (3.0 ng/mL), 29% (0.5 ng/mL), and 4% (0.09 ng/mL), respectively, which were similar to the median urinary concentrations of BPA (3.57 ng/mL) and BPS (0.24 ng/mL) observed in Chinese adults [76]. The abovementioned studies indicate that exposure to bisphenols occurs throughout human development, with increasing exposure levels from infancy to adulthood.

### 3.4. Exposure Patterns of Bisphenols in the Population

Human exposure to bisphenols present in the environment most commonly occurs through food intake, followed by inhalation and direct skin contact with environmental components such as indoor dust, soil, and surface water [80,81]. Bisphenols ingested through food intake are mainly derived from plastic packaging and containers. Environmental factors such as temperature, heat, and acidity can increase the hydrolysis of ester bonds that bind BPA molecules to epoxy resins and polycarbonate. Leakage of BPA is more common in low-pH polycarbonate solutions, whereas higher temperatures increase the leakage of BPA in epoxy resins [82]. Studies have shown that canned tuna is the main source of BPA exposure among Spanish adolescents. In a study including 23 samples of canned legumes from well-known Italian brands, BPA was detected in 91% of samples [83,84]. In another study from Portugal, half of the 30 canned meat samples had a bisphenols concentration of >50 µg/kg, with the daily intake and hazard index exceeding the standards established by the European Food Safety Authority [85]. Consistently, individuals who consume canned foods regularly have higher urinary concentrations of BPA. Controlling the intake of canned foods and takeaway beverages can reduce the urinary concentrations of BPA and BPS by 53.1% and 63.9%, respectively [86,87,88]. In a study conducted from 2017 to 2019, packaging materials for fish, chicken breast meat, and leafy vegetables collected from Canada and South Africa showed a detection rate of 50.0% for BPS in packaged fish and a detection frequency of 66.7% for BPA in packaged vegetables [89]. Additionally, the average concentration of BPA in pork loin under standard conditions has been reported to be 37.03 ± 6.18 ng/g dry weight, which may be attributed to the long-term intake of feed [90]. In addition to food, beverages are an important source of exposure to bisphenols. Various bisphenols have been detected in beer and functional beverages from Italy [91]. In a study on fresh milk samples from 13 major brands in China, at least two types of bisphenols were detected, with an average increase in concentration from 7.1% to 107.1% owing to enzymatic activity [92].

Skin contact is a major exposure route for certain occupational groups, such as workers in thermal paper coating factories, who have urinary BPA concentrations of 1000–1500 ng/mL, and workers in epoxy resin factories, who have a median serum BPA concentration of 18.75 ng/mL [93,94]. Apart from workers directly involved in the manufacturing of epoxy resins and thermal paper, individuals working as cashiers exhibited the highest urinary bisphenol concentrations. The urinary bisphenols concentration is highest among cashiers (1.12 ng/mL), followed by teachers (0.552 ng/mL) [95]. An Asian study focusing on cashiers found that the urinary concentration of BPA increased after a work shift among cashiers who did not wear gloves; however, this phenomenon was not observed among cashiers who wore gloves [96]. Bisphenols are used in the production of school supplies. The detection rates of BPA, BPF, and BPS in school supplies are >80% (with median concentrations of 161, 23.64, and 14.11 ng/g dw, respectively), making them an important route of bisphenol exposure for students and teachers [97].

Furthermore, various textiles contain bisphenols, with the average concentration being 77.8 ng/g and 1.24 ng/g in masks and traditional cotton-made maternity and infant clothing, respectively [98,99]. Significant amounts of bisphenols have been detected in stockings and pantyhose used by women, and the concentration of BPS ingested through direct contact of the skin with pantyhose is as high as 45.9 ng/kg/day [100]. Inhalation of dust in the air is a major route of bisphenol exposure. The detection frequency of bisphenol exceeds 90% in indoor dust samples from China, and the concentration of BPA in the air is highest at 0.137 ng/m^3^ in the United States of America [100,101]. Researchers speculate that open burning is a potential mechanism for the release of bisphenols into the atmosphere [102].

## 4. Toxic Effects of Bisphenols on the Uterus

The uterus is one of the most important organs in the female reproductive system, and is composed of the endometrium, muscle layer, and outer membrane. These structures coordinate with each other to maintain the normal function of the uterus. The endometrium, which is the innermost tissue of the uterus, mainly includes epithelial cells and glandular tissue; it has a rich supply of blood vessels, and plays an important role in the implantation of a fertilized ovum. The muscle layer, which occupies the main part of the uterus, is composed of smooth muscle cells, and its excellent contractility ensures the development and delivery of the fetus. The outer membrane of the uterus is composed of connective and epithelial tissues that protect and support the uterus by interacting with surrounding tissues [1].

### 4.1. Animal Studies

#### 4.1.1. Changes in Uterine Morphology

Although the effects of bisphenols on the structure and function of the human uterus remain elusive, their adverse effects have been elucidated comprehensively in animal models. The appropriate development and maturity of the uterus during puberty are prerequisites for the normal reproductive function of women. Zhang et al. assessed uterine nutrition in adolescent CD-1 mice and found that continuous 10-day exposure to the release of fluorene-9 bisphenol (BHPF) from plastic bottles (at a dose lower than the concentration of BPA without observed adverse reactions) induced endometrial atrophy and a reduction in uterine weight [103]. In our previous study, significant endometrial contraction was observed in adolescent CD-1 mice exposed to BPB or BPAF for 28 days [104]. Excessive endometrial contraction may lead to problems such as pain, excessive bleeding, and regenerative disorders [95]. In addition, exposure to BPB leads to a significant decrease in the thickness of the extrauterine muscle layer and the height of the luminal epithelium, whereas exposure to BPAF leads to dilation of the endometrial glands [104]. Previous studies have suggested that defects in the formation of the uterine glands result in a decrease in fertility [105,106]. The multi-layer differentiation of endometrial epithelial cells in the uterus after exposure to BPAF may serve as a precursor to endometrial hyperplasia [107]. Zhang et al. reported that exposure of Balb/c mice (aged 5–6 weeks) to 5- and 50-mg/kg TBBPA for 14 days increased the uterine index and aggravated pathological damage, such as uterine edema, inflammatory cell infiltration, and endometrial thickening, accompanied by an increase in the number of blood vessels [108]. Similarly, Wang et al. reported that exposure to TCBPA led to uterine edema in >80% of mice [109]. Another study showed that exposure to BPS (20 mg/kg bw/day) and BPF (100 mg/kg bw/day) during adolescence induced uterine growth, which is a marker of estrogen exposure [110]. This gonadotropic effect was also observed in Sprague Dawley rats exposed to BPAF at doses of 100 mg/kg bw/day and 300 mg/kg bw/day [111]. However, exposure to BPAP significantly reduced uterine weight in adolescent CD-1 mice. Xiao et al. conducted GeneBLAzer™ β-lactamase reporter gene assay and found that BPAP exhibited strong anti-estrogenic activity and its uterine gene expression pattern showed an opposite trend to E2, which may partially explain the anti-uterine nutrition in mice [112]. Although the results of previous studies are inconsistent owing to differences in mouse species and exposure doses, the studies have validated that exposure to bisphenols during development causes long-term changes in the physiological features of the uterus, resulting in long-term effects on the normal development and function of the uterus.

#### 4.1.2. Changes in Overall Fertility

##### Estrus Cycle and Intrauterine Implantation

Several studies have indicated that bisphenols pose a threat to the fertility of female mice. Female rodents possess sexual receptivity and fertility during the estrous cycle, and uterine damage during the estrous cycle can lead to endometrial resorption, which affects uterine receptivity and implantation [113]. Exposure to both BPA and BPS can delay puberty and prolong the estrus cycle in female mice, leading to a decrease in the fertilization rate, implantation index, and total number of live offspring [114,115]. Multiple studies have shown that exposure to BPA during pregnancy and the perinatal period can lead to endometrial hypertrophy in adult offspring of Balb/c mice, accompanied by the presence of glandular and stromal endometriotic structures in the adipose tissue surrounding the reproductive tract [116,117]. Berger et al. found that subcutaneous injection of BPA in early pregnancy increased the area of the uterine lumen and the height of the luminal epithelial cell layer in a dose-dependent manner, resulting in impaired intrauterine implantation [118]. In addition, injection of 40-mg/kg BPA into C57BL6 mice during early pregnancy (0.5–3.5 days) has been reported to delay implantation and increase the perinatal mortality risk [119], whereas exposure to 20-mg/kg BPA has been reported to remarkably reduce the number of implantation sites in rats, and these preimplantation defects may be related to the interference of bisphenols with the uterine homeobox A10 (*Hoxa10*) and its downstream hormone receptor signaling pathways [120].

##### Uterine Arteries and Reproductive Circulation

Maternal uterine circulation plays a crucial role in the normal growth and development of the placenta and fetus. In pregnant rats, orally administered BPA (2.5, 25 and 250 µg/kg/day) results in decreased luminal diameter of the uterine arteries and endothelial dysfunction by downregulating the expression of endothelial nitric oxide synthase 3 (NOS3), ERα, and peroxisome proliferator-activated receptor γ (PPARγ), thereby negatively affecting reproductive circulation [121]. Abnormal uteroplacental blood flow limits the normal growth of the placenta, further leading to decreased placental/fetal weight and pregnancy complications [121]. The purpose of uterine spiral artery reconstruction is to optimise the blood supply in the uterus and provide a favorable environment for embryo implantation and normal development [122]. Exposure to a seemingly ‘safe’ dose of BPA (50 µg/kg/day) before and after implantation causes fetal growth restriction by directly affecting uterine spiral artery remodeling [123]. These findings suggest that bisphenols negatively affect the outcomes of pregnancy by targeting uterine arteries specifically.

##### Intergenerational Transmission

The proliferation and differentiation of uterine glandular tissue begin during the embryonic development stage. Disturbances in endocrine signals during this stage may lead to structural and functional abnormalities in adulthood. The negative effects of bisphenols on the reproductive function of female mice are transmitted to the next generation, resulting in adverse reproductive outcomes in offspring. A study showed that compared with control treatment, perinatal exposure to BPA resulted in early puberty and significant thinning of the endometrial epithelium in female offspring, which may be related to TLR4/NF-κB-mediated inflammatory activation and abnormal autophagy induced by the mTOR signaling pathway [124]. Prenatal exposure to BPA decreased the gestational index in the first (F1) and second (F2) generations, significantly delayed the age at vaginal opening in the F3 generation, and impaired fertility in all offspring over time [125]. Martinez et al. found that the interference of BPA with offspring fertility was attributed to the decrease in the number of implantation sites caused by the reduced expression of uterine tight junction (TJ) proteins [126]. Consistent with mammalian models, BPA exposure led to chromosomal synaptic damage and double-strand break repair (DSBR) in nematode Caenorhabditis elegans models, resulting in increased infertility and embryonic mortality rates [127].

#### 4.1.3. Interference with Hormone Levels

Estrogen and progesterone are key factors associated with pregnancy that are mainly responsible for maintaining the balance of hormone levels in the body. Previous studies have shown that weakened signaling of ER and PPAR during pregnancy leads to intrauterine growth restriction and pre-eclampsia [128,129]. Bisphenols as xenestrogens exert adverse effects on reproductive processes by interfering with receptor binding [1]. The progesterone receptor (PR)-heart and neural crest derivatives expressed 2 (HAND2) signaling pathway is inhibited in the uterus of BPA-exposed mice, resulting in abnormal endometrial epithelial proliferation and interstitial differentiation function and a lack of uterine acceptance [130,131]. Perinatal exposure to BPA induces E2-mediated hormone receptor changes in the uterus of older rats, thereby affecting the expression of proteins involved in the differentiation of uterine glands [132]. Postnatal exposure to BPA decreases E2-mediated PR expression in the endometrium of African green monkeys [133]. Exposure to BPA (50 mg/kg/day) from postnatal day (PND) 10 to PND12 can trigger a genomic endoplasmic reticulum response to regulate the expression of estrogen-responsive genes in the uterus of newborns [134]. Administration of BPA at high doses (400 and 600 mg/kg/day) inhibits the expression of ERα and adhesion proteins in the endometrium, leading to adhesion failure during the implantation of blastocysts [135].

Uterine contractility is a key factor affecting fertility. To adapt to fetal growth, the uterus rapidly expands during pregnancy and returns to its normal size after delivery. BPA stimulation alters uterine contractility in rats by interfering with the prostaglandin F2α (PGF2α) and acetylcholine (neural) pathways, which subsequently trigger a series of intracellular events, such as elevated Ca^2+^ levels, resulting in multiple adverse effects on reproduction [136]. Gupta et al. showed that BPA reduced the magnitude and frequency of spontaneous uterine contractions through a nitrergic mechanism in a dose-dependent manner [137].

#### 4.1.4. Uterine Cancer

Studies have shown that the use of environmentally relevant doses of bisphenols may cause severe uterine diseases, including adenomyosis, leiomyoma, atypical hyperplasia, cervical sarcoma, and stromal polyps. Although more in-depth research is warranted to confirm these findings, existing evidence suggests that bisphenols have adverse effects on uterine health and are associated with severe uterine lesions. Li et al. collected uterine leiomyoma and normal uterine tissue samples from 96 patients and conducted primary cell culture. The results of ChIP-seq and RNA-seq confirmed that BPA promoted the proliferation of uterine leiomyoma cells by acting on X-box binding protein 1 (XBP1) and downstream integrin subunit alpha 2 (ITGA2), thereby activating the PI3K/AKT signaling pathway [138]. Similarly, another study showed that BPA was preferentially enriched in uterine leiomyoma and adjacent myometrial tissue samples from patients [139]. In addition, BPA-treated rats exhibit increased myometrial thickness and active cell proliferation, verifying the role of BPA in the growth of UF [139]. Long-term oral administration of low concentrations of BPA (60 µg/kg/day) to adult C57BL6 mice promotes abnormal proliferation and induces morphological changes in the uterine gland epithelium, which is the site of origin of endometrial cancer. These effects may be partly attributed to the epigenetic mechanism underlying HAND2 (an anti-proliferating factor) promoter hypermethylation and the subsequent increase in Fibroblast Growth Factor (FGF) levels and downstream MAPK activation [140]. A study on rats exposed to BPA (25 and 250 μg/kg bw/day) for a long period (PND6–PND365) reported that 57 in the uterine gene expression profile during estrus (PND365) were associated with endometrial cancer. Moreover, the differential expression of 476 homologous human genes was associated with poor overall survival in the TCGA-endometrial cancer cohort [141].

A recent 2-year National Toxicology Program study found that Wistar Han rats exposed to TBBPA developed tumoral and non-tumoral lesions in the uterus that resembled high-grade type 1 tumors in women in terms of morphological and molecular characteristics [142]. Compared with spontaneous uterine cancer, TBBPA-induced lesions exhibited increased mutation rates of tumor protein 53 (*Tp53*) and overexpression of human epidermal growth factor receptor 2 (*Her2*), suggesting an increase in the risk of cancer [143]. A possible molecular mechanism underlying the development of TBBPA-induced uterine epithelial tumors such as adenomas, adenocarcinomas, and malignant mixed Müllerian tumors (MMMTs) is the inhibition of 17β-hydroxysteroid dehydrogenase (HSD17β) and estrogen sulfotransferases that are responsible for the sulfation of estradiol. Reduced sulfation levels lead to increased bioavailability and metabolism of estrogen to oduce active quinone, which eventually leads to DNA damage in the development of uterine tumors [144,145].

Jones et al. showed that the differential expression of target genes in EM lesions after exposure to BPA and BPAF (3, 30 and 90 mg/kg/day) was related to the hormone status of mice [146]. These toxic substances may disrupt the steroid production pathway and the balance of positive and negative feedback in the hypothalamic–gonadal axis, reduce PR levels, and eventually cause uterine lesions. Compared with BPA, BPAF is more likely to promote the growth of endometrial lesions [146]. Studies investigating the long-term effects of prenatal or neonatal BPA exposure on mice have shown that low doses of environmental-related BPA can lead to serious uterine lesions, such as cystic endometrial hyperplasia and squamous metaplasia, adenomyosis, leiomyoma, atypical hyperplasia, cervical sarcoma, and stromal polyps [147]. These lesions may not be evident in the early stages but gradually manifest during an observation period of up to 18 months [147].

#### 4.1.5. Epigenomic Changes

Epigenetic mechanisms, such as DNA methylation, histone modification, and the role of non-coding RNA, regulate gene expression without inducing changes in the DNA sequence [148]. Understanding these mechanisms may improve the understanding of how hetero-estrogens increase susceptibility to hormone-dependent tumors in adulthood. Existing evidence suggests that epigenetic regulation can be modified by the toxic effects of bisphenols on the uterus.

DNA methylation, a widely investigated epigenetic mechanism, usually occurs in CpG islands [149]. Alterations in DNA methylation are closely related to the occurrence and development of many diseases [149]. Continuous exposure to BPA for 1 week during pregnancy significantly induces the expression of the *Dnmt3b* (a methyltransferase) in uterine stromal cells and increases the mRNA and protein expression of *Hoxa1* in uterine tissues in offspring [150]. In addition, Bromer et al. showed that BPA influenced the DNA methylation pattern of *Hoxa10*, decreasing the methylation levels of cytosine–guanine dinucleotides and introns in *Hoxa10* promoters by 53% and 68%, respectively [150]. Reduced methylation in the promoter sequence of *Hoxa10* ERE leads to increased expression of the driver gene and affects the development of mice [150]. The effects of BPA on the uterus are intergenerational. Hiyama et al. reported that exposure to BPA on days 12–16 of pregnancy led to uterine cavity compression in mouse offspring (F2), which may be related to changes in the methylation pattern of *Hoxa10* in the intron region [151].

The molecular mechanisms underlying CpG methylation at the *Hoxa* gene locus are complex and may involve other epigenetic mechanisms, particularly histone modifications that affect the chromatin structure and gene expression regulation. Exposure to bisphenols may disrupt the balance between activated and inhibitory histone markers, further regulating the expression of target genes [152]. Ensuring sufficient endometrial decidualization is a prerequisite for successful embryo implantation. Exposure to BPA downregulates the expression of mixed-lineage leukemia 1 (MLL1) (a histone methyltransferase) and enhancer of zeste homologue 2 (EZH2), inhibits histone-3 lysine-4 trimethylation (*H3K4me3*) and promotes histone-3 lysine-27 trimethylation (*H3K27me3*) at HOXA10, prolactin (PRL), and insulin like growth factor binding protein 1 (IGFBP-1) promoter regions responsible for these two enzymes, consequently damaging the decidualisation of endometrial stromal cells by affecting ER-mediated histone modification [153]. Consistently, a study on Eker rats showed that low-dose BPA exposure (50 ng/kg) increased the levels of inhibitory *H3K27me3* labeling in the developing uterus, inhibited the expression of estrogen-responsive genes in the uterine myometrium, and exhibited a developmental reprogramming pattern opposite to that of genistein [134]. *H3K4me3* at the promoter region is closely related to the expression of ERβ in the endometrium, and overexpression of ERβ is one of the pathogenic factors for EM [154]. A recent study showed that exposure of mice with EM to BPA upregulated the expression of WD repeat domain 5 (WDR5) through the G protein-coupled estrogen receptor (GPER)-mediated PI3K/mTOR signaling pathway, resulting in increased recruitment of the WDR5–tet methylcytosine dioxygenase 2 (TET2) complex to ERβ [154]. In addition, BPA exposure upregulated *H3K4me3* at the promoter and exon regions and inhibited DNA methylation at CpG islands, promoting the growth of EM [154].

N6 methyladenosine (m6A) plays an important regulatory role in multiple biological processes [155]. METTL3, a key factor responsible for abnormal m6A repair and cell self-renewal, is one of the most important methyltransferases involved in RNA methylation. Orally administered BPA (0.5, 5 and 50 mg/kg) in 4-week-old SD rats interferes with the expression of METTL3 in the uterus in a dose-dependent manner. This finding strongly suggests that BPA exerts damaging effects on the reproductive system through RNA methylation [155].

### 4.2. In Vitro Studies

Numerous *in vitro* studies have demonstrated that bisphenols exert adverse effects on the structure and function of uterine cells through various mechanisms, including affecting cell proliferation and apoptosis, activating receptor pathways, and regulating gene expression.

#### 4.2.1. Receptor-Mediated Biological Effects

It is well established that many actions of bisphenols are mediated by the classical hormone receptors, especially ER (ERα/β), PR, and the G protein-coupled receptor (GPCR) [1]. In one study, endometrial stromal fibroblasts (ESFs) derived from hysterectomy specimens were treated with 5–100 μM of BPA for 48 h. The results showed that high doses of BPA significantly induced the overexpression of IGFBP-1 and downregulated the expression of ERα [156]. In addition, BPA promoted the production of IGF-1 and vascular endothelial growth factor (VEGF) by activating the ERα pathway, thereby stimulating the proliferation and growth of leiomyoma cells. This positive association suggests that BPA promotes the growth of leiomyoma cells in a time- and concentration-dependent manner [157]. Exposure to BPS remarkably upregulates the expression of ERβ and venous protein (VIM) in human endometrial epithelial cells (Ishikawa), subsequently promoting cell proliferation and migration [158]. In addition, exposure to 20 μg/mL of BPA reduces the expression of PR, ERα, and the cell proliferation markers KI-67 and CCND2 in human uterine stromal fibroblast cells (HuF), resulting in the disruption of *in vitro* decidualization of uterine stromal fibroblasts [159]. GPER1, as an estrogen membrane receptor, had also been found to bind to BPA and activate the downstream MAPK/ERK/c-fos signaling pathway, thereby stimulating the proliferation of uterine leiomyoma cells [160].

#### 4.2.2. Transcriptional Regulation Mediated Biological Effects

Both intracellular and extracellular signaling molecules regulate gene expression. External components, such as hormones, cytokines, or environmental factors, interact with receptors on cells and trigger a series of signal transduction pathways that eventually affect gene expression. This signal transduction process can regulate the physiological state, stress response, and immune response of cells. Treatment with 1-μM BPA significantly increased ROS production in human endometrial stromal cells (ESCs) [161]. In addition, it increased the expression of MAPK and NF-κB and the release of inflammatory cytokines [161]. These results suggest that BPA affects the biological changes in the endometrium by inducing oxidative stress and inflammatory signals in ESCs [161]. In another study, BPA (10, 10^3^, and 10^5^ nM) downregulated the expression of miR-149 in the ADP-ribosylation factor 6 (ARF6)-TP53-cyclin E2 (CCNE2) pathway in human endometrial cancer RL95-2 cells, thereby interrupting cell cycle arrest and initiating cancer migration and invasion. At the same time, BPA may weaken the hedgehog signaling suppressor of the fused homolog (SUFU) -GLI family zinc finger 3 (GLI3) pathway by upregulating miR-107, interfering with the DNA repair function of cancer cells [162].

Bisphenols can act on the promoter regions of specific genes to activate transcription and translation through downstream signaling pathways, thereby regulating biological processes such as cell cycle, proliferation, differentiation, and apoptosis in uterine cancer. Yu et al. showed that BPA (10^−6^–10 μM) treatment could accelerate the transition of human uterine leiomyoma (ht-UtLM) cells from the G1 to the S phase, induce the overexpression of SOS1 and Grb2d proteins mediated by the MAPKp44/42/ERK1/2 pathway, and upregulate ERα36; and ERα36, a splicing variant of ER, was confirmed to be involved in various crosstalk pathways such as cancer activation and metastasis as an estrogen responsive receptor [163]. Kang et al. analyzed the mRNA expression profiles of leiomyoma cells after 48 and 96 h of BPA exposure, and the results of KEGG enrichment analysis showed that BPA affected the expression of genes related to the PI3K–AKT signaling pathway, ECM-mediated interactions, and focal adhesion (*COL6A1*, *COL15A1*, *FGF7*, *FN1*, *ITGA6*, and *TNC*), all of which are involved in tumor growth [164]. Li et al. showed that exposure to BPA significantly altered the differential expression of 739 genes in uterine leiomyoma cells [165]. These genes were primarily enriched in the cell cycle and PI3K–AKT signaling pathway, and their dysregulation owing to exposure to BPA led to the disruption of cell cycle regulation and an increase in the levels of inflammatory factors and cell proliferation [165]. Wang et al. showed that exposure to BPA enhanced the growth and colony-forming ability of human endometrial cancer cells (RL95-2) in a dose-dependent manner [166]. Given that these effects involve the overexpression of cyclooxygenase-2 (COX-2) and epithelial–mesenchymal transition (EMT), which are closely related to tumor formation, BPA may promote the migratory and invasive abilities of RL95-2 cells [166]. In another study, treatment with BPA, BPS, and BPAF significantly upregulated the expression of pro-fibrotic genes and activated TGF-β-mediated Smad and ERK signaling pathways in a three-dimensional uterine fibroid culture system, revealing a possible mechanism underlying bisphenol-induced fibrosis [167]. Low doses of BPA can promote the migratory and invasive abilities of cervical cancer cells *in vitro* by activating the IKK-β/NF-κB signaling pathway and increasing the expression of metalloproteinase-9 (MMP-9) and fibronectin (FN) [168].

### 4.3. Epidemiological Research

#### 4.3.1. Uterine Fibroids

UFs are the most common monoclonal tumors of the uterine muscle of the reproductive system in women of reproductive age worldwide [138]. Although benign, the incidence rate is as high as 75%, and they are a major source of female reproductive dysfunction. They may induce dysmenorrhea, uterine bleeding, and pelvic discomfort, and increase the risk of adverse pregnancy outcomes such as infertility, recurrent miscarriage, and premature delivery [138]. With more than 200,000 hysterectomies reported each year in the United States, resulting in an estimated cost of USD 5.9 to 34.4 billion, fibroids have become a major public health problem [169,170]. The pathological features of UFs are increased proliferation of disorganized smooth muscle cells, deposition of extracellular matrix such as collagen, fibronectin, and proteoglycan, and increased sensitivity to sex hormone response [171]. It has been found that the occurrence of uterine leiomyoma involves a combination of genetic and non-genetic factors, among which non-genetic factors such as the *in vitro* environment play a crucial role in its pathogenesis [172].

In one study, the incidence of UFs and urinary concentrations of BPA and phthalate were measured in 495 women. Although no significant association between BPA and the probability of fibroid diagnosis was observed, the average BPA levels were significantly higher in women with UFs (2.1 [1.6, 2.8] μg/g) than in those without UFs (1.5 [1.2, 1.7] μg/g) [173]. In a cohort study on women of reproductive age from Algeria, 75% of women with uterine leiomyoma were found to have significantly higher plasma BPA levels than the control women [174].

#### 4.3.2. Endometriosis and Endometrial Carcinoma

EM is a very common estrogen-dependent gynecological inflammatory disease, characterized by heterogeneous lesions of endometrial glands and stroma outside the uterine cavity [175,176]. The prevalence is currently 6–10% in women of reproductive age and 20–30% in women with reproductive dysfunction, with the frequency soaring to 40–60% in women with pain, infertility, or both [177]. The clinical manifestations of EM are pelvic pain, adhesions, and infertility [178]. Due to the high incidence rate and poor treatment prognosis of this kind of disease in women, the exact pathogenesis of this disease is still a mystery today. The most convincing model is the retrograde menstrual hypothesis proposed by Sampson et al., which involves the retrograde implantation of endometrial tissue fragments into the peritoneum and pelvic cavity during menstruation, which proliferates and induces inflammation, leading to the formation of adhesions [177]. The impact of estrogen on the pathophysiology of EM has been widely accepted, and research has confirmed that estrogen can promote lesion growth and survival [179]. Recently, multiple studies have found molecular and histological phenotypes similar to those of ectopic endometrium in fetuses, which may suggest congenital factors that interfere with the growth and development of the uterus [180,181,182].

UCEC is the sixth most common cancer among women worldwide. According to the International Agency for Research on Cancer, 417,000 new cases of UCEC and 97,000 UCEC-related deaths were reported in 2020 [183]. Approximately 80% of UCEC cases are considered estrogen-dependent. Excessive secretion of estrogen leads to complex glandular hyperplasia, abnormal morphology, and precancerous lesions [184]. In addition, the binding of estrogen receptors to non-antagonistic estrogens may interfere with transcription, consequently leading to cancer [185].

Epidemiological studies have suggested a positive relationship between the urinary concentration of bisphenols and the incidence of EM [186]. Peinado et al. evaluated the relationship between EM and the urinary concentrations of BPA, BPS, and BPF in 124 women aged 20–54 years [187]. Among 35 women with EM, all women had BPA, 28.6% of women had BPF, and 11.4% of women had BPS in their urine. Among 89 control women (without EM), 8.5% of women had BPA, 29.9% of women had BPF, and 16.1% of women had BPS in their urine [187]. Cobellis et al. showed that the serum concentrations of BPA and BPB were 2.91 ± 1.74 and 5.15 ± 4.16 ng/mL, respectively, in 58 patients with peritoneal EM of childbearing age. However, neither BPA nor BPB were detected in serum samples from 11 women in the control group [188]. These studies indicate that the presence of BPA, BPS, BPB, and BPF in women may be associated with the occurrence of EM to some extent. The concentrations of free BPA (2.00 ± 0.84 vs. 1.56 ± 0.34 ng/mL), BPA-conjugated compounds (4.33 ± 1.29 vs. 2.98 ± 0.82 ng/mL), and total BPA (6.34 ± 1.38 vs. 4.54 ± 0.65 ng/mL) have been reported to be significantly lower in urine samples from patients with UCEC than in those from patients with benign uterine disease [189]. A case–control study on women at risk of EM indicated that free and conjugated BPA in urine disrupted the hormone balance, leading to aberrant changes in endometrial tissue during the menstrual cycle and helping to establish and maintain a reflux phenomenon in endometrial tissue in patients with EM [190]. A case–control study conducted by Wen et al. showed that the levels of MMP-2 were 175.98 ng/mL and 145.34 ng/mL and those of MMP-9 were 807.41 ng/mL and 750.74 ng/mL in patients with EM and control individuals, respectively [191]. Urinary BPA concentrations were positively associated with MMP-2 and MMP-9 levels, suggesting that BPA increases the risk of peritoneal EM by upregulating the expression of MMP-2 and MMP-9 [191].

#### 4.3.3. Reproductive Disorders

Studies have validated that bisphenols exert adverse effects on reproductive function through endocrine regulation. Ehrlich et al. examined the effectiveness of *in vitro* fertilization (IVF) in 174 women of childbearing age. Exposure to BPA resulted in a reduction of 24% and 27% in the number of IVF oocytes and normally fertilized oocytes, respectively. This finding indicates that high urinary concentrations of BPA significantly inhibit the formation of blastocysts, thereby affecting their reproductive function [192]. A possible explanation for this finding is that the urinary concentration of BPA is negatively associated with the peak level of E2 in serum, which plays a key role in promoting the activation of dormant blastocysts. In another study, the relationship between urinary BPA concentration and implantation failure was observed in 42% of women in the cohort, and the urinary concentration of BPA among these women (3.80–26.48 µg/L) was two times higher than that among control women [193,194]. Cross-sectional studies have indicated that occupational BPA exposure causes changes in the levels of follicle-stimulating hormone (FSH), luteinizing hormone (LH), 17β-estradiol, PRL, and progesterone (PROG) in women, which may partially explain the disruptive effects of BPA on intrauterine implantation [118,195]. Wang et al. evaluated the relationship between pre-pregnancy urinary concentration of BPA and fertility in 700 Chinese couples, and the results showed that the female fertility rate decreased by 13% and the risk of infertility increased by 23% for each unit increase in the urinary concentration of BPA. Women in the highest quartile of urinary BPA levels had a 64% increase in the likelihood of infertility when compared with women in the lowest quartile, with the negative impact being more significant with increasing age [196]. A study based on data from the National Health and Nutrition Survey (NHANES) conducted in 2013–2014 and 2015–2016 reported that among 789 American women of childbearing age, 94% of women had BPA in their urine. In addition, benzophenone-3 (BP-3), BPA and triclosan (TCS) levels were significantly associated with infertility (overall prevalence rate [PR] = 1.13) [197].

#### 4.3.4. Miscarriage

Exposure to bisphenols can increase the risk of miscarriage. In a case study on recurrent miscarriages (RMs) in human patients, urinary BPA concentrations were positively associated with the risk of RMs, compared with women with urinary BPA concentrations of < 0.16 μg/g Cr. Those with urinary BPA concentrations of 0.40–0.93 μg/g Cr and ≥ 0.93 μg/g Cr had odds ratios of 3.91 (95% CI, 1.23–12.45) and 9.34 (95% CI, 3.06–28.44) for persistent RMs, respectively [198]. In addition, conjugated BPA has been significantly associated with an increased risk of miscarriage in women with euploid and aneuploid (OR = 1.83; 95% CI, 1.14–2.96) [199]. These studies above demonstrate the toxic effects of bisphenols on the uterus, which we have summarized in Figure 1.

## 5. Toxic Effects of Bisphenols on the Ovary

The ovary is a female reproductive organ responsible for the production of ova and steroid hormones. The quality and quantity of follicles, which are the main endocrine and reproductive structures of the ovary, determine the female fertility potential and reproductive lifespan. This section reviews the toxic effects of bisphenols on the ovary.

### 5.1. Animal Studies

#### 5.1.1. Changes in Ovarian Morphology

Exposure of mouse ovaries to BPA leads to follicular cystic expansion, decreased granulosa cell (GC) count, reduced abundance of corpus luteum and preovulatory follicles, and increased abundance of atretic follicles and cysts [200,201]. Similar effects, including the presence of atretic follicles, cyst formation, separation of granulosa cells, vascular congestion, and increased thickness of the tunica albuginea, have been observed in rats exposed to BPA [202]. In addition, rat ovaries exposed to BPA for 28 days exhibit evident accumulation of lipid droplets, condensation of GC chromatin, and presence of autophagosomes [203]. These effects have also been observed in mice exposed to BPF [204]. In hens, exposure to BPA significantly reduces productivity, leading to the degeneration of follicles and stromal cells and an increase in the number of atretic follicles [205]. Similarly, exposure to BPA increases the number of atretic follicles in zebrafish ovaries [206]. Follicular atresia and cyst formation induced by BPA resemble the phenotype of PCOS, which may be mediated by alterations in lipid metabolism and steroidogenesis pathways [201]. In sheep, exposure to BPA during follicle development increases the number of multi-oocyte follicles [207]. In rhesus monkeys continuously exposed to BPA, unenclosed oocytes persist in the medullary region, whereas oocytes in secondary and preovulatory follicles are small and do not grow [208]. Exposure to BPA, BPE, and other bisphenols during pregnancy results in suppressed rupture of the ovarian germ cell nest, decreased number of primary and secondary follicles, and pronounced ovarian atrophy in F1 offspring during adulthood [209,210]. In addition, exposure to BPA leads to ovarian atrophy, increased epithelial cell proliferation, and thickening of the body wall in terrestrial animal models, such as earthworms [211].

Furthermore, rats exposed to BPA have significantly low ovarian weight and follicle count and altered composition and number of follicles [212,213]. The adverse effects of BPA on gonadal activity, fertility, hatching rate, and embryo abnormalities in F0 male and female zebrafish persist until the F3 generation, leading to ovarian atrophy [214]. Similarly, in mice, exposure to BPA during pregnancy significantly reduces the relative weight of the ovaries of offspring, indicating that prenatal BPA exposure has transgenerational effects on the female reproductive system [151].

#### 5.1.2. Changes in Meiotic Division

Exposure of female mice to low doses of BPA leads to severe disruptions in meiosis, resulting in a high rate of meiotic failures and chromosomal disjunction disorders. As detected using antibodies against SYCP3 (a crucial component of the synaptonemal complex) and CREST (a marker for centromeres), prolonged exposure to BPA causes an increase in the number of oocytes with abnormal synaptonemal complex formation [215]. Exposure to BPA in utero results in severe defects in normal chromosome pairing (synapsis) and increases meiotic recombination in mouse oocytes undergoing meiosis [216]. In rhesus monkeys, BPA subtly interferes with chromosome separation during the first phase of meiosis [208]. In female fruit flies, exposure to BPA during meiosis increases double-strand breaks and inhibits synaptonemal complex formation in the ovary [217].

In addition to direct exposure to BPA, prenatal exposure affects early-stage oogenesis in developing fetuses, resulting in minor meiotic defects that increase the likelihood of chromosomal abnormalities in offspring during adulthood. Downregulation of genes involved in cell cycle regulation may limit the expansion of the primordial germ cell population in the fetus [218]. In addition to BPA, its analogs BADGE and BPAF delay the initiation of fetal meiosis and induce non-disomic oocytes, accompanied by changes in gene expression and abnormal mRNA splicing of meiosis-related genes [219]. Bisphenols have a greater impact on meiosis. Daily oral administration of 2-μg/kg bisphenols to female mice impairs meiosis and oogenesis in F1 and F2 generations, leading to abnormal follicle development and severe damage to the reproductive system [220].

#### 5.1.3. Autophagy and Apoptosis

BPA induces GC autophagy by increasing the expression of Adenosine 5‘-monophosphate (AMP)-activated protein kinase (AMPK) and 3-Methyladenine (3-MA) [200]. Exposure to BPA and BHPF increases the expression of autophagy- and apoptosis-related genes in mouse ovaries by inducing oxidative stress and DNA damage, thereby affecting ovarian development [204,221]. In one study, exposure of pregnant mice to BPA increased the relative expression of the pro-apoptotic genes *Caspase-7*, *Caspase-9*, and *Bax* while decreasing the relative expression of the anti-apoptotic gene *Bcl-2*, resulting in an increase in follicle numbers in F1 female mice before puberty, significant ovarian atrophy at sexual maturity, and increased apoptosis at both times [210]. Exposure of Drosophila melanogaster to BPA leads to decreased expression of DNA mismatch repair gene (*Mlh1*), promoting follicular apoptosis [217].

#### 5.1.4. Changes in Ovarian Development and Oocyte Maturation

Postnatal exposure to BPA and BPE accelerates puberty and increases postpartum weight [222]. Prepubertal rats exposed to BPA exhibit suppressed ovarian development, with reduced expression of the follicle development-related genes *Figla* and *H1foo* and increased expression of the anti-Müllerian hormone (AMH) [222]. Exposure to BPA in drinking water results in the inhibition of ovarian growth in Euborellia annulipes [223]. Exposure to BPA, BPS, and BPAF delays oocyte maturation in earthworm and zebrafish ovaries [224,225]. As evidenced by the gonadosomatic index (GSI) and histological features of the ovaries of goldfish (Carassius auratus), exposure to BPA decreases the maturity of the ovary, which recovers after discontinuation of BPA treatment owing to a significant decrease in the expression of *sgnrh*, *fshβ*, and *lhβ*, genes related to the hypothalamic–pituitary–gonadal (HPG) axis [226].

#### 5.1.5. Interference with Hormone Levels

Postnatal exposure to BPA, BPE, and BPS increases the levels of steroid hormones in the serum of female animals [222]. The long non-coding RNA *Fhad1os2* is aberrantly expressed in the ovaries of adolescent mice exposed to BPA, and interacts with RUNX3 to affect the synthesis of estrogen and its receptor in GCs, resulting in precocious puberty in adolescent mice [227]. In adult female mice, the protein expression of P450 aromatase and steroidogenic acute regulatory protein (STAR) is significantly low in GCs and ovarian stromal cells [228]. The effects of BPA on hormone production are transgenerational. Prenatal exposure to BPA (F0 mice) results in decreased levels of cytochrome P450 side-chain cleavage, 3β-hydroxysteroid dehydrogenase 1 (3βHSD1), and aromatase cytochrome P450 (P450AROM) mRNA in F1 and F2 mice [229]. These factors are associated with changes in miRNA expression [230].

Exposure of zebrafish to BPA, BPAF, and BHPF leads to overexpression of aromatase, affecting the synthesis of steroid compounds [231,232]. In addition, these bisphenols alter the endogenous cannabinoid levels in the ovary, leading to reproductive dysfunction [233]. Short-term exposure of *Gobiocypris rarus*, a fish species, to BPA significantly increases the levels of estradiol and testosterone, inducing the transcription of 3 beta-hydroxysteroid dehydrogenase (*Hsd3b*) and enhancing the recruitment of ER to the estrogen response element (ERE) in the ovary [234]. These changes are accompanied by increased levels of *H3K4me3*, *H3K9me3*, and *H3K27me3* and decreased methylation levels of *Hsd3b*, which may be involved in the regulation of the steroidogenic genes *Cyp17a1* and *Cyp19a1a* after exposure to BPA [235]. The methylation levels of *Cyp17a1* and *Cyp19a1a* are decreased after short-term exposure to BPA but increased after long-term exposure to BPA [236]. In contrast to BPA-induced acceleration of puberty and early ovarian maturation, BPS decreases the concentration of estradiol in ovarian follicular and oviductal fluids in ewes [237]. However, exposure to 1-μg/L bisphenols leads to an increase in plasma E2 levels and a decrease in testosterone levels in adult female zebrafish, which are associated with the DNA methylation and upregulation of steroidogenic enzymes (including *Cyp11a*, *Cyp17*, and *3βhsd*) in the ovaries of F2 offspring [238]. It is noteworthy that high-dose and long-term exposure to BPA delays puberty and decreases the secretion of ovarian hormones [235,239].

#### 5.1.6. Oxidative Stress

BPA disrupts the ovarian redox balance by activating Keap1 and inhibiting the Nrf2 signaling pathway (Nrf2, NQO1, and HO-1) [205]. In one study, exposure of adult zebrafish to environmentally relevant concentrations of BPA for 30 days promoted the synthesis of reactive oxygen species/reactive nitrogen species (ROS/RNS) and lipid peroxidation in the ovary, thereby reducing the activity of hydrogen peroxide enzyme and exacerbating oxidative stress responses [206]. On the other hand, BPA triggers an increase in the phosphorylation (activation) of p38 MAPK, NF-κB and inducible nitric oxide synthase (*Nos2a*) and the expression of pro-inflammatory cytokines (*Tnf-α* and *Il-1β*) in the ovary, leading to inflammatory reactions [206]. In addition, bisphenols have been reported to induce ovarian lipid peroxidation and oxidative stress in mouse models [240].

#### 5.1.7. Ovarian Cancer

Several studies have suggested that exposure to bisphenols increases the risk of ovarian cancer. In a study, oral administration of BPA and BPB in adult laying hens resulted in lymphocyte and plasma cell infiltration in the ovaries, and affected reproduction, immune function, and expression of genes involved in carcinogenesis [241]. We have previously shown that exposure of adult female mice to bisphenols causes changes in the immune–inflammatory pathway, and the expression of key genes in this pathway, including *Itgam*, *Dhcr7*, *Fdps*, *Hmgcr*, *Hsd11b1*, and *Srd5a1*, is significantly associated with ovarian cancer [203].

### 5.2. In Vitro Studies

#### 5.2.1. Follicle Formation and Development

BPAF and other bisphenols can affect the viability of the human ovarian granulosa tumor cell line (KGN) and the production of angiogenesis-related factors that are associated with follicle development [242]. Additionally, BPA can partially inhibit mouse follicle growth through the AHR pathway and by preventing DNA demethylation at CpG sites in the *Lhx8* gene in oocytes [243,244]. In one study, rat ovaries collected on postnatal day 4 were cultured in a medium containing BPA, leading to a decrease in the number of primary and secondary follicles accompanied by an upregulation of DNA breakage markers and DNA repair genes [245]. In another study involving sheep, bisphenol reduced the cleavage rate and blastocyst formation rate, oocyte maturation rate, and progesterone secretion [246]. Concurrently, bisphenol decreased the expression of PR and AMH genes, while increasing the activation of MAPK1/3, thereby impairing the developmental potential of sheep oocytes [246].

#### 5.2.2. Meiosis

Exposure to BPA alters the meiosis of human ovarian surface epithelial cells (HOSEpiCs) [247]. Specifically, BPA reduces the expression of *p-Erk1*(extracellular regulated protein kinases) and *p-CaMKII* (calcium/calmodulin-dependent protein kinase Ⅱ) in preovulatory follicles; increases the expression of lesions of mismatch repair protein MLH1; results in genomic imprinting disorders and changes in post-translational histone and centromere structure, which may lead to chromosomal misalignment and errors in meiotic division; and alters the expression of genes that potentially impact the health of offspring [248,249,250]. In addition, BPA accelerates the progression of meiosis, leading to the blockade of the transition from meiotic prophase I to prophase II, which is associated with decreased bidirectional communication between cumulus cells and oocytes (COCs), and the regulation of cell cycle protein B1 expression in germinal vesicle breakdown (GVBD) oocytes [251,252]. Another study focusing on the disruption of intercellular communication induced by BPA demonstrated that exposure to 10^−7^-M BPA decreased the expression of Connexin 43 (Cx43) and intercellular gap junction communication in GCs through the estrogen receptor and mitogen-activated protein kinase pathways [253]. In addition, BHPF inhibits the maturation of oocytes *in vitro* by reducing p-MAPK protein levels and destroying meiotic spindles [204].

#### 5.2.3. Autophagy and Apoptosis

The results of *in vitro* and *in vivo* studies on BPA contaminants have shown a high degree of consistency. Exposure of GCs to BPA increases the phosphorylation of AMPK and ULK1 and decreases the phosphorylation of mTOR, whereas inhibition of autophagy via knockout of AMPK or treatment with 3-MA reverses the adverse effects of BPA [254]. Exposure to BPA and BPAF results in the apoptosis of human cumulus cells surrounding oocytes and GCs, leading to ovarian toxicity [255]. This apoptosis is partly mediated through the GPER-dependent activation of the ROS/Ca^2+^/ASK1/JNK signaling pathway, which upregulates apoptosis-related genes and proteins and downregulates anti-apoptotic genes and proteins [256,257]. The other mechanisms underlying this apoptosis involve Ca^2+^ overload, ERβ activation, and the transmembrane protease ADAM17 [258].

#### 5.2.4. Oxidative Damage

Exposure of KGNs to BPA and its analogs increases the levels of intracellular ROS and decreases antioxidant capacity, aggravating the damage caused to major targets of oxidative stress. In addition, exposure of KGN cells to high concentrations of BPA and BPAF leads to ROS-mediated activation of the ASK1 and JNK pathways, accompanied by a significant increase in intracellular Ca^2+^ levels [200,259]. Mouse oocytes exposed to BHPF *in vitro* exhibit an increase in ROS levels [204].

#### 5.2.5. Chromosomal and DNA Damage

Exposure of ovarian cells to BPA results in increased DNA damage, micronucleus frequency, and conventional chromosomal aberrations, which are primarily characterized by chromosomal breaks, gaps, and fragments [260]. In addition, it upregulates the expression of genes related to pairing synapsis (*Smc1β* and *Sycp1*) and double-strand break signaling and repair (*Spo11*, *Rpa*, *H2ax*, and *Mlh1*) in oocytes [261]. Exposure of bovine cumulus–oocyte complexes (COCs) to BPA and bisphenols for 24 h induces abnormalities in spindle fibers and chromosomal misalignment in a non-monotonic manner [262].

#### 5.2.6. Hormone Production

Both BPA and BPAF are detrimental to the survival of ovarian cells and the production of progesterone [263]. Exposure to environmentally relevant levels of BPA disrupts the synthesis of steroid hormones in human ovarian GCs by inhibiting the expression of hormone synthesis-related genes in the FSHR/GS/AC signaling pathway [264]. Studies have shown that BPA exerts endocrine-disrupting effects on human ovarian GCs through the BPA–CTNNB1–FOXL2–CYP19A1–E2 signaling cascade [265]. BPF induces an increase in progesterone secretion, whereas BPA and bisphenol (BPC) induce a decrease in progesterone secretion [266]. Alterations in the progesterone biosynthesis pathway are mediated by PPARγ-dependent activation of epidermal growth factor receptor and ERK1/2, resulting in increased mRNA expression of STAR [267]. Exposure of pig follicle-derived ovarian GCs to low concentrations of BPA stimulates an increase in E2 concentration, whereas exposure to higher concentrations of BPA decreases E2 concentration [268]. Therefore, BPA interferes with reproductive activity by affecting the synthesis of steroids in GCs *in vitro*, which is consistent with the findings of *in vivo* studies. Additionally, bisphenols affect E2 secretion during both early and late stages of follicle development [269].

### 5.3. Epidemiological Research

#### 5.3.1. Polycystic Ovary Syndrome

PCOS is the most common endocrine disorder in women. It is characterized by ovulatory dysfunction and hyperandrogenism of ovarian origin [270]. Population-based studies have shown a significant association between urinary BPA concentrations and polycystic ovarian morphology assessed on ultrasonography in adolescent girls [271]. Compared with healthy women, women with PCOS have higher BPA concentrations in their urine and follicular fluid and higher BPS concentrations in the serum [272]. Urinary BPA concentrations are negatively associated with ovarian steroids and positively associated with androgens [273]. This relationship suggests that BPA inhibits the synthesis of ovarian steroids [274]. Compared with healthy women with low internal BPA levels, women with PCOS exhibit abnormal levels of certain steroids and phospholipids, indicating impaired activity of pathways associated with steroid hormone biosynthesis, arachidonic acid metabolism, linoleic acid metabolism, and phospholipid metabolism [275]. In addition to affecting the production of steroid hormones, BPA affects the quantity of ovarian follicles and diminishes ovarian reserve in patients with PCOS [276].

#### 5.3.2. Poor Ovarian Response

POR typically refers to diminished ovarian reserve or poor response of the ovaries to exogenous gonadotropin (Gn) stimulation. It is characterized by a low number of developing follicles during ovarian stimulation cycles, low peak estradiol levels, high Gn dosage requirements, high cycle cancellation rates, low oocyte yields, and low clinical pregnancy rates [277]. Women with POR who are exposed to bisphenols exhibit higher levels of DNA methylation in cumulus cells, higher concentrations of BPA in the follicular fluid, and lower oocyte quality and clinical pregnancy rates when compared with women not exposed to bisphenols. In addition, differential expression of ICAM-1 and HLA-G has been observed in cumulus cells [278].

#### 5.3.3. Diminished Ovarian Reserve (DOR)

DOR refers to a decline in the number of follicles and the quality of oocytes within the ovaries. It is characterized by symptoms such as menstrual irregularities, oligomenorrhoea, amenorrhea, and infertility, all of which contribute to reduced female fertility. The clinical indicators of DOR include decreased levels of estradiol and AMH and a decreased number of antral follicles (AFCs) [236]. Among women in infertility clinics, higher urinary concentrations of BPA are associated with a lower number of AFCs [279]. In addition, BPA concentrations in the follicular fluid are significantly higher in women with DOR than in healthy women and are associated with AMH and E2 levels in the follicular fluid [236].

#### 5.3.4. Precocious Puberty

Precocious puberty refers to the early onset of physical changes associated with adolescence. In one study involving 110 girls with precocious puberty and 100 healthy girls, BPA was detected in 40.9% of blood samples from girls with precocious puberty and 5% of blood samples from healthy girls, and there was a positive correlation between the concentration of BPA and the volume of the ovaries [280]. These findings are consistent with those of animal studies demonstrating that bisphenols induce early puberty (Figure 2).

## 6. Toxic Effects of Bisphenols on Other Female Reproductive Organs

### 6.1. Bisphenols as Fallopian Tube Poisons

The fallopian tube, as a channel connecting the uterus and ovaries, ensures the delivery of ova, the occurrence of fertilization, and the migration and implantation of fertilized ova, and is crucial for successful conception and pregnancy [281]. Only a limited number of studies have evaluated the impact of exposure to bisphenols on the fallopian tubes. Studies have shown that exposure to high concentrations of BPA (400 mg/kg) before implantation in mice (E0.5–E3.5) significantly increased the expression of endothelial nitric oxide synthase in the fallopian tubes, leading to apoptosis of incubated blastocyst cells and delayed transfer of embryos from the fallopian tubes to the uterus, resulting in embryo implantation failure [120,282]. Progressive proliferative lesions in the fallopian tubes and Wolffian tubes of offspring adult mice could be observed after low-dose BPA during pregnancy [147,283]. In addition, exposure to BPA in late pregnancy (three months before birth) altered the morphology of fetal fallopian tubes in rhesus monkeys, and the proportion of epithelial cells differentiating into ciliated cells significantly increased. Compared with the control group, the BPA-exposed group showed almost no expression of neutron globin in the fallopian tubes, indicating that BPA severely disrupted the secretion function of the fallopian tubes [284]. BPS exerted its endocrine-disrupting effect by damaging the metabolic state of adult ewes, manifested by a significant decrease in steroid hormones in the fallopian tube fluid, especially the content of E2 and its precursor estrogen [237]. BPA and diethylstilbestrol (DES) could also interfere with the development of the fallopian tubes by regulating the expression of genes such as *Wnt*, *Amh*, and forkhead transcription factor (*Foxl2*) [285,286]. In summary, current research confirmed the effects of bisphenols on the morphology, development, and function of the fallopian tubes. However, factors such as species differences, exposure time, and dosage may also affect the damaging effects of bisphenols. Therefore, further research is needed to verify and deeply understand the relevant mechanisms and determine whether they are suitable for women.

### 6.2. Bisphenols as Vaginal Poisons

Some *in vivo* and *in vitro* studies have investigated the effects of exposure to bisphenols on vaginal morphology and function. Exposure to DES in the uterus significantly increased the risk of cervical lesions and vaginal cancer in women, possibly due to DES interfering with the expression of the key protein TRP63/p63 that determined the transition of vaginal duct epithelium to squamous epithelium [287,288]. The layered structure of vaginal squamous epithelium is a complex system, with multiple layers of cells forming a protective barrier for the vagina. DES-induced the expression of IGF-I and Wnt signaling factors in the mouse vagina, disrupting the stratification of vaginal epithelial cells [289]. In addition, exposure to DES during the neonatal period altered the vaginal gene expression of mice in adulthood, with upregulation of cell cycle-related gene p21 expression in the vagina, and continuous downregulation of basic helix–loop–helix (bHLH) transcription factor genes (*Hes1*, *Hey1*, and *Heyl*) expression, manifested as excessive epithelial cell proliferation and inhibition of stromal cell proliferation [290,291]. Laronda et al. used a mouse model to confirm that downregulation of RUNX family transcription factor 1 (RUNX1) and inhibition of the BMP/Activin SMAD signaling pathway were key molecular mechanisms underlying DES-induced vaginal adenosis and even clear cell adenocarcinoma [292].

Ahmed et al. examined the morphological changes of the vagina after BPA exposure in rats, and the results showed that 0.1 mg/kg BPA treatment reduced the upper layer of the local vaginal plaque, whereas 50 mg/kg BPA directly caused the foamy cytoplasmic vacuolation and the pyknotic nuclei of the epithelial nucleus in the vagina, which might further lead to vaginal tumor lesions [293]. Both BPA and BPF exposure could alter the secretion function of vaginal epithelial cells in mice and lead to increased vaginal mucus [294,295]. Zhang et al. found that oral BPA (5 and 50 mg/kg) disrupted the normal growth pattern of vaginal epithelial cells, resulting in shorter estrus duration in the exposed group, and that it had been shown to interfere with uterine and ovarian development by participating in the methylation of METTL3 [296]. Exposure to low-dose BPS (0–100 μg/kg) induced blastocyst apoptosis by altering oxidative stress and hormone levels in the reproductive tract, as well as N2f1/P53 dependent pathways [297]. The structure and function of the vagina are regulated by various factors, and current research is mostly focused on BPA and DES. Further in-depth analysis of the damaging effects of other bisphenol analogs on the vagina, including hormones, cell growth, and differentiation, should be conducted.

## 7. Conclusions

There are various types of bisphenols in the environment, and their endocrine disrupting mechanisms on the female reproductive system are very complex. Taking into account all endpoints listed in this review, some epidemiological studies have supported the relationship between bisphenols and female reproductive system diseases; in vivo and *in vitro* studies have elucidated the key molecular mechanisms underlying the adverse effects of bisphenols on female reproductive health. Bisphenols destroy the structure and function of the reproductive organs of exposed individuals, induce reproductive system diseases, and affect key reproductive processes such as the estrus cycle, ovum production, and intrauterine implantation, eventually leading to reproductive dysfunction. Bisphenols exert these adverse effects by interfering with biological processes such as hormone production, oxidative stress, transcription, and epigenetic regulation of key factors. In addition, prenatal exposure to bisphenols may induce developmental reproductive disorders and increase disease susceptibility in offspring through intergenerational effects. It should be mentioned that this review includes a large number of adverse outcome indicators for the female reproductive system and different methods for measuring these indicators. Therefore, integrating these practical outcomes means that we provide a narrative review of the survey results. Due to the influence of various confounding factors such as exposure time, dose, and exposure subjects on the included report outcomes, we did not compare the reproductive toxicity strength among various bisphenols. In the future, we will further screen the included literature and conduct a meta-analysis to obtain the statistical relationship between bisphenols and female reproductive system damage.

## 8. Challenges and Perspectives

Based on the evidence presented in this review, a consensus has been formed on the adverse effects of bisphenols on female reproductive health. At the same time, the current challenges between bisphenols and female reproductive damage have been identified, and these knowledge gaps need to be filled to better assess the actual risk level of female exposure to bisphenols.

The safe dose of bisphenol exposure remains unknown. According to the EPA, the no-observed-adverse-effect level (NOAEL) of BPA is 5 mg/kg/day, and the lowest-observed-adverse-effect level (LOAEL) is 50 mg/kg/day [146]. In 2015, the European Food Safety Authority (EFSA) established a temporary tolerable daily intake (TDI) of 4 μg/kg bw/day for BPA. Subsequently, the CEP established a TDI of 0.2 ng/kg bw/day for BPA [298]. To date, most toxicological studies have reported the use of high doses and concentrations of bisphenols. The findings of these studies may be insufficient in reflecting the actual adverse effects of bisphenols on relevant reproductive and endocrine endpoints in women. Currently, some scholars have developed conversion methods regarding the human exposure dose and animals. Extrapolating animal-to-human dose requires considerations of body surface area, pharmacokinetics, and physiological time. Currently, four different methods are described in the literature, namely dose factor method, similar drug method, pharmacokinetic-guided method, and comparative method for dose assessment. There is now a relatively comprehensive system available for calculating equivalent doses [299]. Therefore, in future studies focusing on reproductive endpoints in female animals, bisphenols should be used at doses consistent with human exposure to obtain more accurate data.

At present, toxicological risk assessment focuses more on the toxic effects of individual chemicals. Rappaport et al. indicated that more than 1000 small molecules can be simultaneously detected in human blood [300]. It is noteworthy that the actual exposure to bisphenols in daily life is not limited to a single compound. In addition, considering the effects of complex ‘chemical mixtures’ on human health is important for simulating the actual exposure. Liu et al. proposed a ‘leave-one-out’ method based on high-throughput metabolomics to examine the toxic effects of 23 mixed exogenous pollutants on breast cancer cells and assess the relative contribution of each pollutant to the synergistic toxic effects [301]. Future studies should pay more attention to the synergistic toxic effects of mixed EDCs on reproductive health and continue to search for and establish models and methods related to actual human exposure, such as high-throughput omic techniques and organoid cultures.

To date, most studies have focused on the effects of BPA on female reproductive health. However, bisphenols are metabolized in the body through enzymes and pathways such as sulfation, methylation, or glucosylation, and the products vary based on the species, dosage, and exposure time. Assessing the reproductive toxicity of bisphenols and their metabolites remains a major challenge. With the rapid development of machine learning, deep learning, and artificial intelligence, future studies can use these techniques to analyze existing experimental data to elucidate the relationship between bisphenols and reproductive toxicity. The establishment of predictive models may help to screen and identify the structural characteristics and mechanisms through which bisphenols induce potential reproductive toxicity. This strategy may reduce reliance on large numbers of experiments, improve research efficiency, and guide the design and development of health- and environmentally-friendly alternatives.

Some studies have discussed the synergistic effects of different bisphenols on reproductive health. However, extensive interactions and connections exist among various organs in the human body, and the endocrine system is regulated by hormones to maintain internal homeostasis. The endocrine system is composed of glands that secrete chemical messengers (hormones) that interact with specific targets (receptors), leading to a range of functional regulation [302]. At present, molecular docking and molecular dynamics simulations have captured from the perspective of computational biology that bisphenol substances bind to the ligand binding domains of ER, GPER, and TR with different energies and conformations, and interfere with the transcription of downstream genes [303,304,305,306]. Further work is needed to incorporate other hormones (such as peptide, protein, and monoamine biosynthesis, release, and metabolism) and receptors (such as membrane progesterone receptors and neurotransmitter receptors) that combine endogenous hormone levels with exposure concentrations within bisphenols, taking into account the crosstalk effects throughout the developmental context of the body.

Existing evidence suggests that serum or urinary concentrations of bisphenols are associated with female reproductive system diseases. However, in most studies, these concentrations have been measured using serum or urine samples collected at a single time point, which may not accurately explain the effects of long-term bisphenol exposure on disease progression. In the future, more epidemiological studies should be conducted using advanced surveillance methods to assess the effects of long-term bisphenol exposure in the general population and evaluate the in-population exposure levels of pollutants. In addition, race, age, reproductive characteristics (menstrual cycle, parity, and menopause) and long-term exposure sources (housing history and water sources) should be considered to elucidate the adverse effects of bisphenols on female reproductive health at different levels. Altogether, a combination of actual exposure levels, epidemiological surveys, and toxicological assessment may guide the development of public health policies and corresponding risk management strategies to ensure the safety of daily environmental exposure and food supply chains.

## Figures and Tables

**Figure 1 toxics-11-01000-f001:**
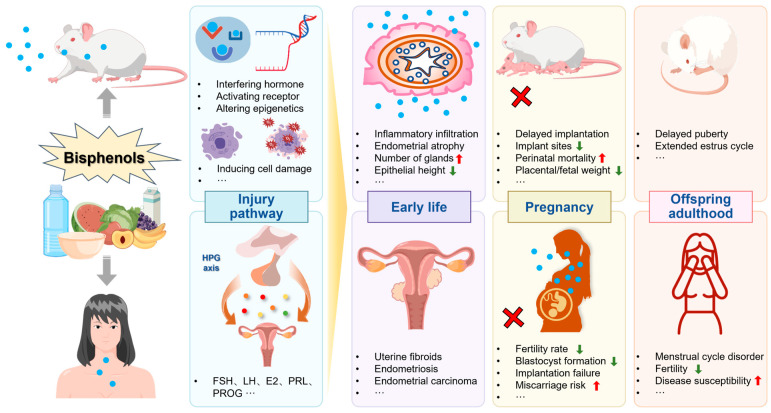
The toxic effects of bisphenols on the uterus. A schematic diagram shows the damage to the uterus and its injury pathway caused by exposure to bisphenols at different periods. HPG axis, hypothalamic–pituitary–gonadal axis; FSH, follicle-stimulating hormone; LH, luteinizing hormone; E2, estradiol; PRL, prolactin; PROG, progesterone. The blue dots in the figure represent bisphenols, the red arrows represent upregulation, and the green arrows represent downregulation. Figure created with Generic Diagramming Platform (Available online: https://gdp.renlab.cn/, China, accessed on 26 November 2023, Copyright license serial number: RJCR2023IKDNE9) and Figdraw (Available online: https://www.home-for-researchers.com/static/index.html#/, China, accessed on 26 November 2023, Copyright license ID: ARAOS85b88 and IASYU8c3b8).

**Figure 2 toxics-11-01000-f002:**
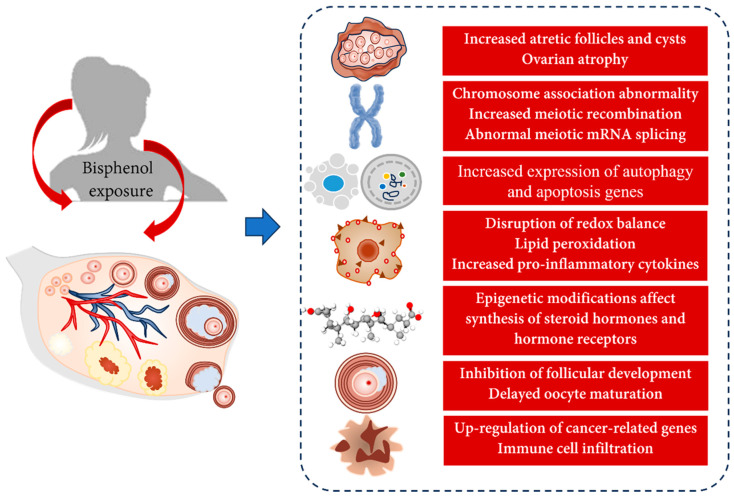
Toxic effects of bisphenols on the ovaries. The schematic shows the damage to the ovary caused by different periods of exposure to bisphenols, including: pathological changes, DNA and chromosome damage, autophagy and apoptosis, oxidative stress, hormone-receptor binding, ovarian development, and ovarian tumors. Red arrows: Effects of bisphenol exposure on the ovary, blue arrows: specific details about the mechanisms.

**Table 2 toxics-11-01000-t002:** Concentrations of bisphenols detected in human samples.

Study Population	Country	Sample Size	Measurement Time	Sample Sources	Detection Period	BPA		BPS		BPF		BPB		TBBPA		Reference
						Conc. (Mean/Median)	DR. (%)	Conc. (Mean/Median)	DR. (%)	Conc. (Mean/Median)	DR. (%)	Conc. (Mean/Median)	DR. (%)	Conc. (Mean/Median)	DR. (%)	
Pregnant women	Netherlands	1213	2016	Urine	Early pregnancy	1.67 ng/mL	78.8	0.35 ng/mL	68.1	0.58 ng/mL	40.4					Philips EM et al., 2019 [59]
					Mid-pregnancy	1.46 ng/mL	93.3	0.24 ng/mL	29.1							
Pregnant women	South Korea	196	2017–2019	Urine	Pregnancy	2.10 ng/mL	96.2	0.10 ng/mL	62.1	0.20 ng/mL	84.4					Kim S et al., 2021 [61]
Pregnant women	China	2023	2015–2018	Serum	Pregnancy	2.03 ng/mL	99	0.09 ng/mL	86.9	0.44 ng/mL	61.8	0.24 ng/mL	89	0.52 ng/mL	66.1	Liang J et al., 2020 [67]
Pregnant women	South Africa	60	NF	Serum	Pregnancy	1.16 ng/mL										Gounden V et al., 2021 [65]
Infants					Infancy	0.75 ng/mL										
Pregnant women	America	635	2010	Urine	Early pregnancy	0.68 ng/mL	74	0.11 ng/mL	52							Gaylord A et al., 2023 [66]
					Mid-pregnancy	0.43 ng/mL	58	0.13 ng/mL	57							
					Late pregnancy	0.73 ng/mL	73	0.09 ng/mL	51							
Children	South Korea	619	2012–2018	Urine	4 years old	3.29 μg/g creatinine										Kim JI et al., 2022 [68]
					6 years old	2.36 μg/g creatinine		0.10 μg/g creatinine		0.20 μg/g creatinine						
					8 years old	1.96 μg/g creatinine		0.16 μg/g creatinine		0.40 μg/g creatinine						
Children	Netherlands	471	NF	Urine	6 years old	2.50 nmol/L	73.8	0.13 nmol/L	25.2							Silva CCV et al., 2021 [69]
Children	Slovenian	246	2016–2019	Urine	6–9 years old	2.10 ng/mL		0.30 ng/mL		0.11 ng/mL						Tkalec Ž et al., 2020 [70]
Adolescents					11–15 years old	1.90 ng/mL	96.2	0.36 ng/mL		0.17 ng/mL						
Children	Egypt	97	NF	Urine	3–8 years old	1.16 ng/mL										Youssef MM et al., 2018 [71]
Children	China	345	2018–2019	Serum	6–11 years old	1.60 ng/mL	63	0.043 ng/mL	43	0.075 ng/ml	68					Guo C et al., 2021 [72]
Adolescents	China	1317	2013–2016	Serum	6–19 years old	1.2 ng/mL	97.8	0.3 ng/mL	88.4	0.2 ng/mL	54.8					Wang Y et al., 2021 [73]
Adolescents	Belgium	423	2016–2020	Urine	14–15 years old	1.05 ng/mL	86	0.12 ng/mL	83	0.14 ng/mL	97					Gys C et al., 2021 [74]
Adults	America	1046	2013–2016	Urine	>12 years old	1.23 ng/mL	96.8	0.58 ng/mL	92.3	0.41 ng/mL	56.5					Choi JY et al., 2021 [75]
	South Korea	3268	2015–2017		>19 years old	1.27 ng/mL	98.1	0.03 ng/mL	54.8	0.10 ng/mL	40.7					
Adults	Norway	144	2016–2017	Urine	24–72 years old	3.0 ng/mL	96	0.5 ng/mL	29	0.09 ng/mL	4					Husøy T et al., 2019 [76]
Adults	China	160	2018	Urine	19–25 years old	4.89 ng/mL	99	0.15 ng/mL	88	0.13 ng/mL	80					Zhang H et al., 2020 [14]

Conc. mean: mean concentration; Conc. median: median concentration; DR: detection rate; NF: not shown in the references.

## Data Availability

Not applicable.

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
