# Peer review of "Invisible Hand behind Female Reproductive Disorders: Bisphenols, Recent Evidence and Future Perspectives"

_toxics, 2023, doi:10.3390/toxics11121000_

Round 1

Reviewer 1 Report

Comments and Suggestions for Authors

This review by Wu et al. aims to give an up-to-date status on how environmental bisphenols can affect female reproduction and related disorders. The authors conclude that epidemiological studies have confirmed a relationship between bisphenols and female reproductive diseases and that in vitro and in vivo studies have elucidated the key molecular mechanism which results in the destruction of reproduction organs. This topic is relevant, as exposure to endocrine disruptors, including BPA, is prevalent, and their effects on human health are debated. A strength of the review is that it comprehensively covers the potential negative aspects of exposure and gives a detailed overview of current exposure levels and routes. 

However, the review, as currently written, is persistently one-sided and does not reflect the existing debate. For example, the review does not reflect that the leading cause for concern regarding bisphenol exposure is in-utero exposure, children, and possibly adolescents (and also this is debated due to the low concentrations of exposure) and that the theoretical underpinnings for impact on fertile women are questionable. (Stronger evidence exists for a cause-effect of decreased fertility in males). On this note, the authors do not delve sufficiently into the molecular mechanism (incl. BPA mimicking estrogen) and do not discuss that women’s higher estrogen levels would outcompete BPA in the binding to estrogen receptors. The authors do not discuss how the proposed effects of BPA exposure compare to the effects of DES (another synthetic estrogen that was given to pregnant women and impacted girls in utero). Many effects found in animal studies used experiments with very high doses while humans have a low-dose exposure; this distinction should be made clearer. On this note, it is essential to put the levels of exposure in relation to the endogenous levels of estrogens (endogenous estrogens are, for example, considerably higher in women than in female mice). In addition, numerous incorrect statements are made (exemplified below). Also, the flow and structure of the review should be improved.

To be noted, several reviews on similar topics have been published recently (incl. Interdonato L, et al. Endocrine Disruptor Compounds in Environment: Focus on Women's Reproductive Health and Endometriosis. Int J Mol Sci. 2023 Mar 16;24(6):5682; Srnovršnik T, Virant-Klun I, Pinter B. Polycystic Ovary Syndrome and Endocrine Disruptors (Bisphenols, Parabens, and Triclosan)-A Systematic Review. Life (Basel). 2023 Jan 4;13(1):138Raszewski G, Jamka K, Bojar H, Kania G. Endocrine disrupting micropollutants in water and their effects on human fertility and fecundity. Ann Agric Environ Med. 2022 Dec 27;29(4):477-482Stavridis K, et al. Bisphenol-A and Female Fertility: An Update of Existing Epidemiological Studies. J Clin Med. 2022 Dec 5;11(23):7227.).

Specific comments 

Frontpage figure: This figure does not have a logical flow and seems to include incorrect information. It is unclear why the ovary and the uterus exposure would differ (different exposure figures).  The exposure subpanels also do not show how bisphenols can interact with the nuclear estrogen receptors (the panel top left seems to indicate that receptors (looks like antibodies) on cell surfaces bind the bisphenols – but this is inaccurate; the receptors are primarily located in the nucleus). The structures of bisphenols also lack the most common one (BPA).

The review should include the official gene symbols for all genes mentioned throughout.

Figure 1: The legend states that molecular mechanism is shown, but it is not. It simply states “activating receptors”, and the figure has some of the same inconsistencies as the front-page figure. 

Line 46: This sentence (that BPA has profound effects on reproduction, metabolism, immune system, etc) needs references, as well as critical reflection.

Line 59: “... mechanism through which bisphenols cause damage to the reproductive system”. This is one example of how this review (throughout) does not reflect the uncertainty of current evidence nor the divergent conclusions drawn in the field. 

Line 72. “... remarkably increases amino acids in the mother”. This sentence needs to be more specific. Do the authors mean higher amino acid levels in serum? 

Line 106. “Exposure to BPA ... beginning in infancy.” The most critical exposure is likely to begin in utero.

Line 141 (line 220, and throughout): Information in tables does not need to be wholly repeated in the text.

Line 181: “..low doses of bisphenols in the environment lead to their accumulation in human bodies”. This statement is either inaccurate or needs to be specified. BPA is readily metabolized and secreted within hours in humans (may differ in fetuses).

Line 536 (section Non-coding RNAs): This section is misplaced. It would make sense to describe what is known about the mechanisms first (incl. binding to estrogen receptors and initiating transcription of both mRNA and non-coding RNAs in the same way as other estrogens. And what is known of other mechanisms. On this note, it would be valuable to state which mediator of BPA effects is expressed (or not expressed) in the named cell lines (such as RL95-2 here)

Line 553-554: “The binding of bisphenols to the receptors on cell membranes”. This is incorrect. The ERs that the statement refers to (the authors should specify if they refer to ERalpha (ESR1) and ERbeta (ESR2) or both) are located in the cell nucleus. Later, the authors mention GPR30 (the official gene symbol GPER1 should be mentioned), which is a transmembrane receptor, but this is not clarified in the text. 

Line 569-570: This statement (“acting as a specific DNA sequence”) does not make sense. GPR30 is (as mentioned in the comment above) a transmembrane receptor, not a DNA sequence (and neither does it bind to DNA sequences).

Line 577: Here, octylphenol and nonylphenol are suddenly mentioned. What is the reason for this? (If included, this needs to be introduced). Also, 5 micromolar is not a low dose. It is exceptionally high. 

Line 584: Similar to the above, the 1 micromolar BPA is exceptionally high (compared to human exposure), this should be clarified, and line 592 again refers to 10 micromolar as low.

Linre595: Here, ERa36 suddenly appears. This a splice variant of ERa, but this must be explained and put into context, or the reader will not understand.

Comments on the Quality of English Language

The English is overall of high level. 

Author Response

Journal: Toxics

Manuscript ID: toxics-2732595

Title: Invisible Pusher of Female Reproductive Disorders: Bisphenols, Recent Evidence and Future Perspectives

Dear Prof,

We thank you and the reviewers for your careful consideration of our paper and your very helpful comments and recommendations. We have considered current comments and incorporated the reviewers' suggestions into the manuscript to make a further improvement on it. (Please see the point-by-point response to the reviewers’ comments).

Overall, we have made improvements that we trust address your and the reviewers’ requests and concerns. Hopefully, our manuscript has been strengthened and it is now acceptable for publication in Toxics.

Sincerely,

Huifeng Yue, Ph.D.

The point-by-point response to the Editor and Reviewers’ comments

Reviewer Comments for the Author

This review by Wu et al. aims to give an up-to-date status on how environmental bisphenols can affect female reproduction and related disorders. The authors conclude that epidemiological studies have confirmed a relationship between bisphenols and female reproductive diseases and that in vitro and in vivo studies have elucidated the key molecular mechanism which results in the destruction of reproduction organs. This topic is relevant, as exposure to endocrine disruptors, including BPA, is prevalent, and their effects on human health are debated. A strength of the review is that it comprehensively covers the potential negative aspects of exposure and gives a detailed overview of current exposure levels and routes.

However, the review, as currently written, is persistently one-sided and does not reflect the existing debate. For example, the review does not reflect that the leading cause for concern regarding bisphenol exposure is in-utero exposure, children, and possibly adolescents (and also this is debated due to the low concentrations of exposure) and that the theoretical underpinnings for impact on fertile women are questionable. (Stronger evidence exists for a cause-effect of decreased fertility in males). On this note, the authors do not delve sufficiently into the molecular mechanism (incl. BPA mimicking estrogen) and do not discuss that women’s higher estrogen levels would outcompete BPA in the binding to estrogen receptors. The authors do not discuss how the proposed effects of BPA exposure compare to the effects of DES (another synthetic estrogen that was given to pregnant women and impacted girls in utero). Many effects found in animal studies used experiments with very high doses while humans have a low-dose exposure; this distinction should be made clearer. On this note, it is essential to put the levels of exposure in relation to the endogenous levels of estrogens (endogenous estrogens are, for example, considerably higher in women than in female mice). In addition, numerous incorrect statements are made (exemplified below). Also, the flow and structure of the review should be improved.

To be noted, several reviews on similar topics have been published recently (incl. Interdonato L, et al. Endocrine Disruptor Compounds in Environment: Focus on Women's Reproductive Health and Endometriosis. Int J Mol Sci. 2023 Mar 16;24(6):5682; Srnovršnik T, Virant-Klun I, Pinter B. Polycystic Ovary Syndrome and Endocrine Disruptors (Bisphenols, Parabens, and Triclosan)-A Systematic Review. Life (Basel). 2023 Jan 4;13(1):138; Raszewski G, Jamka K, Bojar H, Kania G. Endocrine disrupting micropollutants in water and their effects on human fertility and fecundity. Ann Agric Environ Med. 2022 Dec 27;29(4):477-482; Stavridis K, et al. Bisphenol-A and Female Fertility: An Update of Existing Epidemiological Studies. J Clin Med. 2022 Dec 5;11(23):7227.).

Authors:

We thank the reviewer for the encouraging comments and helpful suggestions. We carefully addressed comment by the reviewer.

We reviewed the literature to include a large number of indicators of adverse outcomes in the female reproductive system and different methods of measuring these indicators. Therefore, integrating these actual results means that we provide a narrative review of the survey results. The fetal origin hypothesis of adult diseases suggests that the risk factors of exposure to the intrauterine environment can affect fetal development during the sensitive window period and increase the risk of specific diseases in offspring during childhood and adulthood. In addition, considering that many endocrine disorders increase with age, we have extended the concept of bisphenols damage to the female reproductive system to the entire life cycle.

In the section of “Bisphenols contamination in humans”, we list the risks of exposure to bisphenols at various life stages. 1) Studies on bisphenol exposure during pregnancy comprise the largest portion of all population-based data. We have summarised the levels of internal exposure to bisphenols in pregnant women and foetuses in various countries in lines 195-231 of the article. We found that maternal bisphenol exposure can cross the placental barrier into the foetal bloodstream, thereby interfering with initial reproductive development. Specifically described as " In a study including 1,213 pregnant women from the Netherlands (2016), the urinary concentrations of ……indicating that it can be transferred from the mother to the fetus through the placental barrier ". 2) In addition, we have assessed bisphenol exposure in children from different countries in lines 246-258. We found that serum concentrations of bisphenols were generally higher in children than in infancy and foetal life. 3) In adolescents, contrary to what we might expect, the levels of exposure to bisphenols were slightly lower than in childhood, but the effects of bisphenols on adolescents should not be ignored as puberty is a critical window for reproductive development. This section is discussed in the article " In a study including 1,317 adolescents from……observed in children and infants." in lines 259-267. 4) Bisphenol exposure in adults is more likely to be due to social and occupational factors. Population-based cohorts in China, the United States, and South Korea showed the highest levels of internal exposure in adults (lines 268-278), suggesting that the risk of human exposure to bisphenol substances is long-standing and slowly increasing.

Epidemiological data confirmed the association of bisphenols with precocious puberty and menstrual cycle disturbances in adolescents (lines 973-978), reproductive system diseases in women of childbearing age (lines 621-688 and lines 939-952), and reproductive disorders (lines 690-715 and lines 954-971) and an increased risk of miscarriage in pregnant women (lines 717-724).

Experimental animal models have demonstrated a causal relationship between exposure and disease phenotypes, including the effects of bisphenol substances on rodents themselves, pregnancy, and the entire life stage of offspring puberty. The damage to the body is manifested by the destruction of its morphological structure (lines 342-374 and lines 739-767) and changes in hormone levels (lines 426-451 and lines 814-828) and severe diseases of the reproductive system (lines 453-499 and lines 855-862), the main effects on adolescent mice were delayed puberty and impaired development (lines 380-382, lines 801-812 and lines 865-876). The main effects on pregnant mice were decreased overall fertility and impaired reproductive circulation (lines 377-408, lines 761-767 and lines 779-788), which may also be transmitted to offspring through intergenerational effects, resulting in delayed reproductive development and reproductive disorders in offspring (lines 410-425, lines 762-765, lines 786-788 and lines 820-824). In vitro experiments revealed the molecular mechanism of germ cell damage and even carcinogenesis caused by exposure at the cellular level (lines 557-618 and lines 864-936). It is important to note that because differences in animal testing methods can make it difficult to draw broad conclusions, we must be careful to extrapolate causality to humans.

The endocrine system is composed of glands that secrete chemical messengers (hormones) that interact with specific targets (receptors), leading to a range of functional regulation. We reviewed literature on the damage of bisphenols to the reproductive system by interfering with hormone levels (lines 427-451, lines 814-842 and lines 922-936). At present, molecular docking and molecular dynamics simulations have captured from the perspective of computational biology that bisphenol substances bind to the ligand binding domains of ER, GPER, and TR with different energies and con-formations, and interfere with the transcription of downstream genes (lines 1117-1122). Based on this, we have included relevant literature on the molecular mechanisms mediated by bisphenol mimicking hormone binding receptors (lines 557-573). In addition, we have provided additional information on the estrogeni, non-estrogenic, anti-adrogenic properties of bisphenols (lines 66-78). The correlation between endogenous hormone levels and bisphenol levels in serum/urine was elucidated in the included population studies (lines 693-703, lines 943-951 and lines 968-970). Further work is needed to incorporate other hormones (such as peptide, protein, and monoamine biosynthesis, release, and metabolism) and receptors (such as membrane progesterone receptors, neurotransmitter receptors) that combine endogenous hormone levels with exposure concentrations within bisphenols, taking into account the crosstalk effects throughout the developmental context of the body (lines 1123-1127).

Due to the influence of various confounding factors such as exposure time, dose, and exposure subjects on the included report outcomes, we did not compare the reproductive toxicity strength among various bisphenols. In the future, we will further screen the included literature and conduct a meta-analysis to obtain the statistical relationship between bisphenols and female reproductive system damage (lines 1061-1065).

We explained the current standard for exposure dose of bisphenols in lines 1073-1081 of the revised manuscript. The high doses used in animal exposure are not the actual doses exposed to humans. The number of pollutant exposure studies using animal experiments is huge, but the tolerance of the same pollutant varies greatly between humans and animals, so it is necessary to convert human exposure doses to animal exposure doses. In order to better equate animal exposure levels with human exposure levels, researchers have experimentally derived methods for converting drug doses between humans and animals and between various types of animals [1] (lines 1081-1086). Specifically, the conversion can be done as follows:

The dose by factor method applies an exponent for body surface area (0.67), which account for difference in metabolic rate, to convert doses between animals and humans. Thus, human equivalent dose (HED) is determined by the equation:

HED (mg / kg = Animal NOAEL mg/kg) × (Weight animal [kg]/Weight human [kg]) (1–0.67) (1)

Dose is equally related to body weight although it is not the lone factor which influences the scaling for dose calculation. The correction factor (Km) is estimated by dividing the average body weight (kg) of species to its body surface area (m2) The Km factor values of various animal species (Table 1) is used to estimate the HED as:

HED (mg / kg) = Animal does (mg / kg) × (Animal Km / Human Km) (2)

As the Km factor for each species is constant, the Km ratio is used to simplify calculations. Hence, Equation 2 is modified as:

HED (mg / kg) = Animal does (mg / kg) × Km ratio (3)

The Km ratio values provided in Table 1 is easily obtained by dividing human Km factor by animal Km factor or vice versa. For instance, the Km ratio values for rat is 6.2 and 0.162, obtained by dividing 37 (human Km factor) by 6 (animal Km factor) and vice versa, respectively. Thus, usually to obtain the HED values (mg/kg), one can either divide or multiply the animal dose (mg/kg) by the Km ratio provided in Table 1.

Table 1. Human equivalent dose calculation based on body surface area

Species

Reference body weight(kg)

Working weight range (kg)

Body surface

Area (m2)

To convert dose in mg/m2, multiply by Km

To convert animal dose in mg/kg to HED in mg/kg, either

Divide animal dose by

Multiply animal dose by

Human

60

-

1.62

37

-

-

Mouse

0.02

0.011-0.034

0.007

3

12.3

0.081

Hamster

0.08

0.047-0.157

0.016

5

7.4

0.135

Rat

0.15

0.08-0.27

0.025

6

6.2

0.162

Ferret

0.30

0.16-0.54

0.043

7

5.3

0.189

Guinea pig

0.40

0.208-0.700

0.05

8

4.6

0.216

Rabbit

1.8

0.90-3.0

0.15

12

3.1

0.324

Dog

10

5-17

0.5

20

1.8

0.541

Monkeys

3

1.4-4.9

0.25

12

3.1

0.324

Marmoset

0.35

0.14-0.72

0.06

6

6.2

0.162

Squirrel monkey

0.60

0.29-0.97

0.09

7

5.3

0.189

Baboon

12

7-23

0.60

20

1.8

0.541

Micro pig

20

10-33

0.74

27

1.4

0.730

Mini pig

40

25-64

1.14

35

1.1

0.946

Therefore, in future studies focusing on reproductive endpoints in female animals, bisphenols should be used at doses consistent with human exposure to obtain more ac-curate data.

Reference:

[1] Nair AB, Jacob S. A simple practice guide for dose conversion between animals and human. J Basic Clin Pharm. 2016, 7(2): p. 27-31.

Response to comment

Reviewer:

  1. Frontpage figure: This figure does not have a logical flow and seems to include incorrect information. It is unclear why the ovary and the uterus exposure would differ (different exposure figures). The exposure subpanels also do not show how bisphenols can interact with the nuclear estrogen receptors (the panel top left seems to indicate that receptors (looks like antibodies) on cell surfaces bind the bisphenols – but this is inaccurate; the receptors are primarily located in the nucleus). The structures of bisphenols also lack the most common one (BPA).

Authors:

We thank for reviewer’s careful review and valuable comments. We have revised the incorrect subpanels in the frontpage figure and added annotations to illustrate the different damages of bisphenols to the uterus and ovaries. The structure of BPA has been added to the “structure” section.

Reviewer:

  1. The review should include the official gene symbols for all genes mentioned throughout.

Authors:

We thank for reviewer’s reminder, we examined the manuscript carefully and used official gene symbols and abbreviations when genes first appeared, and abbreviations were used in the rest of the manuscript.

Reviewer:

  1. Figure 1: The legend states that molecular mechanism is shown, but it is not. It simply states “activating receptors”, and the figure has some of the same inconsistencies as the front-page figure.

Authors:

We appreciate your constructive comments. We have revised the incorrect subpanels in the Figure 1 and added annotations to illustrate the damage pathways of the bisphenols shown in the schematic diagram and the resulting early life, pregnancy, and offspring uterine injuries. In addition, the captions have also been changed accordingly.

Reviewer:

  1. Line 46: This sentence (that BPA has profound effects on reproduction, metabolism, immune system, etc) needs references, as well as critical reflection.

Authors:

We appreciate the reviewer’s comments. According to the comments, we have supplemented the reference in lines 46-50 in revised manuscript. In addition, we provided detailed information to illustrate that BPA and its analogs has profound effects on reproduction, metabolism, immune system in lines 86-99 of the revised manuscript.

Reviewer:

  1. Line 59: “... mechanism through which bisphenols cause damage to the reproductive system”. This is one example of how this review (throughout) does not reflect the uncertainty of current evidence nor the divergent conclusions drawn in the field.

Authors:

We appreciate your constructive comments and observations. After a long search and summary of the “Bisphenols and Reproduction Literature”, we found no evidence of beneficial effects of bisphenols on the reproductive system, and both population and in vitro and in vivo experimental data indicate varying degrees of adverse effects on the reproductive system. The uncertainty of the evidence is also explored in the conclusion and perspective sections of the article, such as “However, bisphenols are metabolised in the body through enzymes and pathways such as sulfation, methylation or glucosylation, and the products vary based on the species, dosage and exposure time. Assessing the reproductive toxicity of bisphenols and their metabolites remains a major challenge. With the rapid development of ma-chine learning, deep learning and artificial intelligence, future studies can use these techniques to analyse existing experimental data to elucidate the relationship between bisphenols and reproductive toxicity. Establishment of predictive models may help to screen and identify the structural characteristics and mechanisms through which bisphenols induce potential reproductive toxicity. This strategy may reduce reliance on large numbers of experiments, improve research efficiency and guide the design and development of health- and environment-friendly alternatives.” in lines 1005-1115.

Reviewer:

  1. Line 72. “... remarkably increases amino acids in the mother”. This sentence needs to be more specific. Do the authors mean higher amino acid levels in serum?

Authors:

Thanks very much for the helpful suggestion. We further explained in detail the amino acid levels in lines 92-95 of the revised manuscript, and we revised it to “Exposure to BPB and BPAF leads to accelerated differentiation of the mammary glands (MGs), and in addition, non-targeted metabolomics results show significantly elevated amino acid levels within the maternal MGs, which can lead to amino acid imbalance or hyperammonemia in newborns.”

Reviewer:

  1. Line 106. “Exposure to BPA ... beginning in infancy.” The most critical exposure is likely to begin in utero.

Authors:

We thank for reviewer’s careful review and very critical advice. We apologise for our misrepresentation and we have amended the sentence as “Since BPA is mainly used in everyday products, exposure to BPA during infancy is unavoidable”, in lines 128-129 in the revised article.

Reviewer:

  1. Line 141 (line 220, and throughout): Information in tables does not need to be wholly repeated in the text.

Authors:

We are grateful for the reviewer's careful review and valuable comments, which we believe are very necessary, and we have made bibliographic cuts in lines 162-191, and lines 197-280, and throughout the text so that there is as little duplication as possible between the table and the content of the article.

Reviewer:

  1. Line 181: “..low doses of bisphenols in the environment lead to their accumulation in human bodies”. This statement is either inaccurate or needs to be specified. BPA is readily metabolized and secreted within hours in humans (may differ in fetuses).

Authors:

We appreciate the reviewer’s careful review and valuable comments. We have revised the sentence in lines 197-199 to read: "Progressively higher levels of bisphenol contaminants in the environment also significantly increase the risk of human exposure." to avoid misrepresentation.

Reviewer:

  1. Line 536 (section Non-coding RNAs): This section is misplaced. It would make sense to describe what is known about the mechanisms first (incl. binding to estrogen receptors and initiating transcription of both mRNA and non-coding RNAs in the same way as other estrogens. And what is known of other mechanisms. On this note, it would be valuable to state which mediator of BPA effects is expressed (or not expressed) in the named cell lines (such as RL95-2 here)

Authors:

We agree and appreciate the reviewer’s comments. We have placed this section on lines 585-591 of the revised manuscript and provided a detailed explanation that the mechanism of BPA on human endometrial cancer RL95-2 cells was mediated by transcription regulatory factors, which initiated cancer cell proliferation and impaired DNA repair, we revised it to “In another study, BPA (10, 103, and 105 nM) downregulated the expression of miR-149 in the ADP-ribosylation factor 6 (ARF6)-TP53-cyclin E2 (CCNE2) pathway in human endometrial cancer RL95-2 cells, thereby interrupting cell cycle arrest and initiating cancer migration and invasion. At the same time, BPA may weaken the hedgehog signaling suppressor of fused homolog (SUFU) -GLI family zinc finger 3 (GLI3) pathway by up-regulating miR-107, interfering with the DNA repair function of cancer cells”.

Reviewer:

  1. Line 553-554: “The binding of bisphenols to the receptors on cell membranes”. This is incorrect. The ERs that the statement refers to (the authors should specify if they refer to ERalpha (ESR1) and ERbeta (ESR2) or both) are located in the cell nucleus. Later, the authors mention GPR30 (the official gene symbol GPER1 should be mentioned), which is a transmembrane receptor, but this is not clarified in the text.

Authors:

We thank for reviewer’s reminder. We apologize for the misunderstanding caused by our inaccurate statement, and we revised it to “It is well-established that many actions of bisphenols are mediated by the classical hormone receptors, especially ER (ERα/β), PR and G protein-coupled receptor (GPCR) [1]” in lines 559-560 of the revised manuscript. We carefully examined the ER mentioned throughout the text and designated it as ERα or ERβ. In lines 573-575 of the revised manuscript, we changed GPR30 to GPER1, and we have stated that GPER1 is an oestrogen membrane receptor.

Reviewer:

  1. Line 569-570: This statement (“acting as a specific DNA sequence”) does not make sense. GPR30 is (as mentioned in the comment above) a transmembrane receptor, not a DNA sequence (and neither does it bind to DNA sequences).

Authors:

We appreciate the reviewer’s comments. According to the recommendation, we revised it to “GPER1, as an oestrogen membrane receptor, had also been found to bind to BPA and activate the downstream MAPK/ERK/c-fos signaling pathway, thereby stimulating the proliferation of uterine leiomyoma cells” in lines 573-575 of the revised manuscript.

Reviewer:

  1. Line 577: Here, octylphenol and nonylphenol are suddenly mentioned. What is the reason for this? (If included, this needs to be introduced). Also, 5 micromolar is not a low dose. It is exceptionally high.

Authors:

We thank the reviewer for this comment. Octylphenol (OP) and nonylphenol (NP) are non-ionic surfactants used widely as emulsifiers, detergents, solubilisers, antistatic agents, and dispersing agents in domestic, industrial and agricultural products, they are also used in plastic additives and pesticides. We have re-searched the structural formulas of NP and OP, which are outside of the bisphenols we are concerned about, so we have deleted the literature on NP and OP included here. In addition, we have also corrected the statement that 5 μM is a low dose.

Reviewer:

  1. Line 584: Similar to the above, the 1 micromolar BPA is exceptionally high (compared to human exposure), this should be clarified, and line 592 again refers to 10 micromolar as low.

Authors:

Thanks very much for the helpful suggestion. We explained the current standard for exposure dose of bisphenols in lines 1075-1083 of the revised manuscript. Indeed, compared to human exposure, 1 μM and 10 μM are not low doses. Therefore, we have removed inappropriate wording from the manuscript.

Reviewer:

  1. Line595: Here, ERa36 suddenly appears. This a splice variant of ERa, but this must be explained and put into context, or the reader will not understand.

Authors:

We agree and appreciate the reviewer’s comments. We have added an explanation of ERα36 in lines 598-600 of the revised manuscript, we revised it to “ERα36, a splicing variant of ER, had been confirmed to be involved in various crosstalk pathways such as cancer activation and metastasis as an oestrogen responsive receptor”.

Reviewer 2 Report

Comments and Suggestions for Authors

This is a very well-prepared and written elaboration on bisphenols and their impact on female reproduction. It will have a lot of readers due to nicely composed and presented literature information from the best publications both clinical and experimental.

-line 69 neurotransmitter circulation- which exactly?
- BPA or bisphenols should be indicated in the subject

-the conclusion should also highlight the limitations of the current research

-KGN cells and other similar should be explained

-The estrogeni, non-estrogenic, anti-adrogenic nature of BPA should be highlighted

Author Response

Journal: Toxics

Manuscript ID: toxics-2732595

Title: Invisible Pusher of Female Reproductive Disorders: Bisphenols, Recent Evidence and Future Perspectives

Dear Prof,

We thank you and the reviewers for your careful consideration of our paper and your very helpful comments and recommendations. We have considered current comments and incorporated the reviewers' suggestions into the manuscript to make a further improvement on it. (Please see the point-by-point response to the reviewers’ comments).

Overall, we have made improvements that we trust address your and the reviewers’ requests and concerns. Hopefully, our manuscript has been strengthened and it is now acceptable for publication in Toxics.

Sincerely,

Huifeng Yue, Ph.D.

The point-by-point response to the Editor and Reviewers’ comments

Reviewer Comments for the Author

This is a very well-prepared and written elaboration on bisphenols and their impact on female reproduction. It will have a lot of readers due to nicely composed and presented literature information from the best publications both clinical and experimental.

Authors:

We thank for editor and reviewer’s carefully revision. We answered the following concerns raised by the reviewers in detail.

Response to comment

Reviewer:

  1. line 69 neurotransmitter circulation- which exactly?

Authors:

We appreciate the reviewer for this comment. Following the reviewer’s advice, we further explained the neurotransmitter circulation, that is, dopamine and serotonin. In lines 88-91 of the Introduction section in revised manuscript, we added “Administration of BPA/BPS to animals during pregnancy may lead to an increase in the proportion of dopamine+ trophoblast giant cells (GCs), while the proportion of serotonin+ GCs correspondingly decreases, which may affect the placental-brain axis of the developing fetus.”

Reviewer:

  1. BPA or bisphenols should be indicated in the subject

Authors:

We thank the reviewer for this comment, we have distinguished “BPA” and “bisphenols” used in the paper.

Reviewer:

  1. the conclusion should also highlight the limitations of the current research

Authors:

We agree and thank the reviewer for this comment. In the Conclusion section, we pointed out the limitations of this review, we added “It should be mentioned that this review includes a large number of adverse outcome indicators for the female reproductive system and different methods for measuring these indicators. Therefore, integrating these practical outcomes means that we provide a narrative review of the survey results. Due to the influence of various confounding factors such as exposure time, dose, and exposure subjects on the included report out-comes, we did not compare the reproductive toxicity strength among various bisphenols. In the future, we will further screen the included literature and conduct a meta-analysis to obtain the statistical relationship between bisphenols and female reproductive system damage” in lines 1059-1066 of the revised manuscript.

In addition, we further elaborated on the limitations in current research in lines 1074-1139 of the Challenge and Perspective section in revised manuscript.

In lines 1075-1083 we cited the EPA and the European Food Safety Authority (EFSA) standards for BPA, detailing that a limitation of current research is that most toxicological studies have reported the use of high doses and concentrations of bisphenols. The results of these studies may be insufficient in reflecting the actual adverse effects of bisphenols on relevant reproductive and endocrine endpoints in women.

In lines 1091-1096, we pointed out that current toxicological risk assessments focused more on the toxic effects of individual chemicals. However, in daily life, the actual exposure to bisphenols in daily life is not limited to a single compound.

In lines 1104-1105, we explained that the limitation of the current research is that it did not take into account the metabolites of bisphenols in vivo and their corresponding reproductive toxicity.

In lines 1115-1118, we believe that the limitation of current research is that it did not consider the extensive interactions and connections between bisphenols and the endocrine system.

In lines 1129-1133, we explain that the limitation of current epidemiological studies is that the internal exposure levels of the population are measured using serum or urine samples collected at a single time point, which may not accurately explain the impact of long-term exposure to bisphenols on disease progression.

Reviewer:

  1. KGN cells and other similar should be explained

Authors:

We agree and thank for reviewer’s careful review, this is an oversight in our work, we have explained the KGN cells, as well as other similar cases have been revised, respectively, in line 930, line 55, line 68, line 130, line 265, line 462, line 463, line 530, line 543 and line 568.

Reviewer:

  1. The estrogeni, non-estrogenic, anti-adrogenic nature of BPA should be highlighted

Authors:

We thank for reviewer’s valuable comments and careful review, we have added the section on BPA having estrogenic and antiandrogenic activity in lines 66-78. The additions are “With the gradual research on BPA, it has been found that its chemical structure is similar to that of estradiol, and it possesses oestrogenic and antiandrogenic biological activities [1]. It acts as a weak agonist binding to oestrogen receptors (ER) α and β as well as androgen receptors, resulting in endocrine disrupting effects [2], as evidenced by Sohoni P et al.'s anti-androgen screening experiments using Saccharomyces cerevisiae [3]. These effects can be carried over to the offspring, e.g., exposure to BPA during gestation in rats causes damage to epithelial cell proliferation levels, androgen receptor expression, and prostate structure in male offspring [4]. It has been found that BPA competes with E2 for binding to ER in an in vitro cell proliferation assay of the human breast cancer cell line MCF-7, but their estrogenic effects are counteracted in the pres-ence of ER antagonists [5]. All of the above studies have shown that BPA produces oestrogenic and anti-androgenic bioactivities through binding to hormone receptors, affecting the normal function of the body's own oestrogenic or androgenic actions.”

Reference:

[1] Hart, RJ. The Impact of Prenatal Exposure to Bisphenol A on Male Reproductive Function. Front Endocrinol (Lausanne), 2020. 11: 320.

[2] Sonnenschein, C. and Soto, AM., An updated review of environmental estrogen and androgen mimics and antagonists. J Steroid Biochem Mol Biol, 1998. 65(1-6): 143-50.

[3] Sohoni, P. and Sumpter, JP., Several environmental oestrogens are also anti-androgens. J Endocrinol, 1998. 158(3):327-39.

[4] Bernardo, BD., et al., Genistein reduces the noxious effects of in utero bisphenol A exposure on the rat prostate gland at weaning and in adulthood. Food Chem Toxicol, 2015. 84: 64-73.

[5] Stroheker, T., et al., Steroid activities comparison of natural and food wrap compounds in human breast cancer cell lines. Food Chem Toxicol, 2004. 42(6): 887-97.

Reviewer 3 Report

Comments and Suggestions for Authors

Review: Invisible Pusher of Female Reproductive Disorders: Bisphenols, Recent Evidence and Future Perspectives

Authors present a comprehensive review of current literature focused on Bisphenols and their link with female reproductive disorders.

While review is well citied, more care should be taken so citations are placed in text closer to where that are first referenced (preferability at the end of the sentence they are first referenced).

Questions

Lines 53-58: outline high dietary intake of BPS, BPF and BPA.  But all the citations are from studies in China, do other countries have similar levels? please highlight this is an issue everywhere

Line 95: "search for articles focusing on the contamination status, sources and exposure routes of bisphenols published between 2018 and 2023."

Why limit your search to articles published after 2018?

Line 295: "The urinary bisphenols concentration is highest among cashiers (1.12 ng/mL)"

Question: in previous sentence you list two other professions with higher urinary concentrations?

Suggestions

Line 2: "Pusher" is not correct English. 

Suggest change "Invisible Pusher of Female Reproductive Disorders" to "Invisible Determinant of Female Reproductive Disorders"

Line 12: Suggest change "The increased exposure to bisphenols has raised..." to

"The increased exposure to bisphenols, a chemical used in large quantities for the production of polycarbonate plastics, has raised..."

Line 28: Image at bottom of page lacks figure legend.  Does not appear to add anything to the understanding of the review.  Suggest delete this image.

Line 42: "endocrine-disrupting chemicals (EDCs) can mimic or block natural hormone activity in the human body."

Citation(s) needed

Line 46: "As an EDC, bisphenol A (BPA) has profound adverse effects"

Should briefly mention in introduction what/why BPA exposure is so common (e.g. a chemical used in large quantities for the production of polycarbonate plastics) before outlining its dangers

Line 46: "(BPA) has profound adverse effects on reproduction, metabolism, development and nervous and immune systems."

Citation(s) needed

Line 63: Suggest change "Gal et al." to "Ziv-Gal et al. "

Lines 68-74: "Administration of BPA/BPS to animals during pregnancy may cause damage to the giant cell neurotransmitter circulation and immune reactivity of placental trophoblasts. Exposure to BPB and BPAF leads to accelerated differentiation of the mammary glands and remarkably increases amino acid levels in the mother. The imbalance of placental and breast functions leads to abnormal foetal development and altered expression of genes related to carcino-genesis in the adult offspring [14-16]."

Please add citations at end of sentences, instead of waiting to group them all at end of paragraph.

Lines 174;219 Tables 1, 2: Reduce line spacing so tables appear on 1 page.

Line 723 Figure 1: Figure 1 is first mentioned in line 324, but appears at line 723.  Figure 1 should be moved so it's closer to when first mentioned in text.

Line 335: " in adolescent CD-1 mice exposed to BPB or BPAF for 28 days."

Citation(s) needed

Line 339: " exposure to BPAF leads to dilation of endometrial glands. "

Citation(s) needed

Line 348: "during adolescence induced uterine growth, which is a marker of oestrogen exposure. "

Citation(s) needed

Line 351: "However, exposure to BPAP significantly reduced uterine weight in adolescent CD-1 mice. "

Citation(s) needed

Line 372: "In addition, injection of 40-mg/kg BPA into C57BL6 mice during early pregnancy (0.5– 3.5 days) has been reported to delay implantation and increase the perinatal mortality risk, whereas exposure to 20-mg/kg BPA has been reported to remarkably reduce the number of implantation sites in rats. "

Citation(s) needed

Line 382: "In pregnant rats, orally administered BPA (2.5, 25 and 250 μg/kg/day) results in decreased luminal diameter of uterine arteries and endothelial dysfunction by downregulating the expression of endothelial nitric oxide synthase 3 (NOS3), estrogen receptor α (ERα), and peroxisome proliferator-activated receptor γ (PPARγ), thereby negatively affecting reproductive circulation. "

Citation(s) needed

Line 416: "Bisphenols as xenoestrogens exert adverse effects on reproductive processes by interfering with receptor binding."

Citation(s) needed

Line 421: "Perinatal exposure to BPA induces E2-mediated hormone receptor changes in the uterus of older rats, thereby affecting the expression of proteins in- volved in the differentiation of uterine glands. "

Citation(s) needed

Line 447: "another study showed that BPA was preferentially enriched in uterine leiomyoma and adjacent myometrial tissue samples from patients."

Citation(s) needed

Line 461: "rats exposed to TBBPA developed tumoral and non-tumoral lesions in the uterus that resembled high-grade type 1 tumours in women in terms of morphological and molecular characteristics. "

Citation(s) needed

Line 474: "Jones et al. showed that the differential expression of target genes in EM lesions after exposure to BPA and BPAF (3, 30 and 90 mg/kg/day) was related to the hormone status of mice. "

Citation(s) needed

Line 481: "can lead to serious uterine lesions, such as cystic endometrial hyperplasia and squamous metaplasia, adenomyosis, leiomyoma, atypical hyperplasia, cervical sarcoma and stromal polyps."

Citation(s) needed

Line 490: Suggest change "epigenetic regulation can reveal the toxic effects of bisphenols" to "epigenetic regulation can be modified by the toxic effects of bisphenols"

Line 493: "DNA methylation, a widely investigated epigenetic mechanism, usually occurs in CpG islands. "

Citation(s) needed

Line 495: "Continuous exposure to BPA for 1 week during pregnancy significantly induces the expression of the Dnmt3b (a methyltransferase) in uterine stromal cells and increases the mRNA and protein expression of Hoxa1 in uterine tissues in offspring."

Citation(s) needed

Line 501: "Taylor et al. showed that BPA influenced the DNA methylation pattern of Hoxa10. The effects of BPA on the uterus are intergenerational. "

Citation(s) needed

Line 521: "H3K4me3 at the promoter region is closely related to the expression of ERβ in the endometrium. "

Citation(s) needed

Line 522: "A recent study showed that exposure of mice with EM to BPA upregulated the expression of WD repeat domain 5 (WDR5) through the G protein-coupled oestrogen receptor (GPER)-mediated PI3K/mTOR signalling pathway, resulting in increased recruitment of the WDR5–tetmethylcytosine dioxygenase 2 (TET2) complex to ERβ. "

Citation(s) needed

Line 529: "N6 methyladenosine (m6A) plays an important regulatory role in multiple biological processes. "

Citation(s) needed

Line 532: "Orally administered BPA (0.5, 5 and 50 mg/kg) in 4-week-old SD rats interferes with the expression of METTL3 in the uterus in a dose-dependent manner."

Citation(s) needed

Line 540: "Treatment with BPA (10, 103 and 105 nM) induces the overexpression of miR-107 in the hedgehog signalling pathway and downregulates miR-149 and its related DNA repair genes in the ADP-ribosylation factor 6 (ARF6)-TP53-cyclin E2 (CCNE2) pathway in human endometrial cancer RL95-2 cells. "

Citation(s) needed

Lines 555; 631; 792; 845; 856; 871: Suggest change "In a study" to "In one study"

Line 577: "Ko et al. found that low-dose (5 μM) treatment with octylphenol (OP) and nonylphenol (NP) significantly reduced the ability of human endometrial cells to secrete PRL and IGFBP1. "

Citation(s) needed

Line 579: "Increased expression of the bone morphogenetic protein 2 (BMP2), a key protein involved in decidualization, induced the expression of Wnt4 and controlled the expression of forkhead box O1 (FOXO1) and its downstream gene left-right determination factor 1 (LEFTY1) through the β-catenin pathway. "

Citation(s) needed

Line 584: "In another study, treatment with 1-μM BPA significantly increased ROS production in human endometrial stromal cells (ESCs)."

Citation(s) needed

Line 596: "Kang et al. analysed the mRNA expression profiles of leiomyoma cells after 48 and 96 h of BPA exposure. "

Citation(s) needed

Line 600: "Li et al. showed that exposure to BPA significantly altered the differential expression of 739 genes in uterine leiomyoma cells."

Citation(s) needed

Line 605: "Wang et al. showed that exposure to BPA enhanced the growth and colony-forming ability of human endometrial cancer cells (RL95-2) in a dose-dependent manner. "

Citation(s) needed

Line 618: "UF is the most common monoclonal tumor of the uterine muscle of the reproductive system in women of reproductive age worldwide. "

Citation(s) needed

Line 631: "the incidence of UFs and urinary concentrations of BPA and phthalate were measured in 495 women."

Citation(s) needed

Line 648: "Sampson et al."

Citation(s) needed.  No Sampson et al found in reference list.

Line 664: "Peinado et al. evaluated the relationship between EM and the urinary concentrations of BPA, BPS and BPF in 124 women aged 20–54 years."

Citation(s) needed

Line 669: "Cobellis et al. showed that the serum concentrations of BPA and BPB were 2.91 ± 1.74 and 5.15 ± 4.16 ng/mL, respectively, in 58 patients with peritoneal EM of childbearing age. "

Citation(s) needed

Line 681: "A case–control study conducted by Wen et al. showed that the levels of MMP-2 were 175.98 ng/mL and 145.34 ng/mL and those of MMP-9 were 807.41 ng/mL and 750.74 ng/mL in patients with EM and control individuals, respectively."

Citation(s) needed

Line 689: "Ehrlich et al. examined the effectiveness of in vitro fertilisation (IVF) in 174 women of childbearing age. "

Citation(s) needed

Line 703: "Wang et al. evaluated the relationship between pre-pregnancy urinary concentration of BPA and fertility in 700 Chinese couples. "

Citation(s) needed

Line 715: "In a case study on recurrent miscarriages (RMs) in human patients, urinary BPA concentrations were positively associated with the risk of RMs."

Citation(s) needed

Line 986 Figure 2: Figure 2 is first mentioned in line 733, but appears at line 986.  Figure 2 should be moved so it's closer to when first mentioned in text.

Line 739: "Similar effects, including the presence of atretic follicles, cyst formation, separation of granulosa cells, vascular congestion and increased thickness of the tunica albuginea, have been observed in rats exposed to BPA. "

Citation(s) needed

Line 792: "exposure of pregnant mice to BPA increased the relative expression of the pro-apoptotic genes Caspase-7, Caspase-9 and Bax while decreasing the relative expression of the anti-apoptotic gene Bcl-2, resulting in an increase in the ovarian apoptotic rate in F1 female mice before puberty and adulthood. "

Citation(s) needed

Line 814: "The long non-coding RNA Fhad1os2 is aberrantly expressed in the ovaries of adolescent mice exposed to BPA. "

Citation(s) needed

Line 820: "Prenatal exposure to BPA (F0 mice) results in decreased levels of cytochrome P450 side-chain cleavage, 3β-hydroxysteroid dehydrogenase 1 (3βHSD1), and aromatase cytochrome P450 (P450AROM) mRNA in F1 and F2 mice."

Citation(s) needed

Line 837: "However, exposure to 1-μg/L bisphenols leads to an increase in plasma E2 levels and a decrease in testosterone levels in adult female zebrafish. "

Citation(s) needed

Line 845: "exposure of adult zebrafish to environmentally relevant concentrations of BPA for 30 days promoted the synthesis of reactive oxygen species/reactive nitrogen species (ROS/RNS) and lipid peroxidation in the ovary, thereby reducing the activity of hydrogen peroxide enzyme and exacerbating oxidative stress responses."

Citation(s) needed

Line 856: "oral administration of BPA and BPB in adult laying hens resulted in lymphocyte and plasma cell infiltration in the ovaries. "

Citation(s) needed

Line 871: "rat ovaries were collected on postnatal day 4 and cultured in a medium containing BPA. "

Citation(s) needed

Line 874: "in vitro maturation of primordial germ cell–oocyte complexes with or without bisphenols showed that bisphenols decreased the rates of cleavage and blastocyst formation when compared with the control group."

Citation(s) needed

Line 959: "POR typically refers to diminished ovarian reserve or poor response of the ovaries to exogenous gonadotropin (Gn) stimulation. It is characterised by a low number of developing follicles during ovarian stimulation cycles, low peak oestradiol levels, high Gn dosage requirements, high cycle cancellation rates, low oocyte yields and low clinical pregnancy rates."

Citation(s) needed

Line 979: "In a study involving 110 girls with precocious puberty and 100 healthy girls, BPA was detected in 40.9% of blood samples from girls with precocious puberty and 5% of blood samples from healthy girls."

Citation(s) needed

Author Response

Journal: Toxics

Manuscript ID: toxics-2732595

Title: Invisible Pusher of Female Reproductive Disorders: Bisphenols, Recent Evidence and Future Perspectives

Dear Prof,

We thank you and the reviewers for your careful consideration of our paper and your very helpful comments and recommendations. We have considered current comments and incorporated the reviewers' suggestions into the manuscript to make a further improvement on it. (Please see the point-by-point response to the reviewers’ comments).

Overall, we have made improvements that we trust address your and the reviewers’ requests and concerns. Hopefully, our manuscript has been strengthened and it is now acceptable for publication in Toxics.

Sincerely,

Huifeng Yue, Ph.D.

The point-by-point response to the Editor and Reviewers’ comments

Reviewer Comments for the Author

Authors present a comprehensive review of current literature focused on Bisphenols and their link with female reproductive disorders.

While review is well citied, more care should be taken so citations are placed in text closer to where that are first referenced (preferability at the end of the sentence they are first referenced).

Authors:

We thank for editor and reviewer’s carefully revision. We answered the following concerns raised by the reviewers in detail.

Response to comment

Reviewer:

  1. Lines 53-58: outline high dietary intake of BPS, BPF and BPA. But all the citations are from studies in China, do other countries have similar levels? please highlight this is an issue everywhere

Authors:

We thank for reviewer’s careful review and valuable comments, I have added some BPA daily intake for other national populations in the article, in lines 57-61. The revised section reads “Similarly, the daily intake of BPA for Turkish women was 170 ng/kg bw/day based on 24-hour dietary review data [1], while Canadian pregnant women had a lower average daily BPA intake at 32 ng/kg bw/day [2]. The BPA intake for the Portuguese population was between these two values, with adolescents and adults having a daily intake of 79.1 and 46.1 ng/kg bw/day, respectively [3].”

Reference:

[1] Çiftçi, S., et al., Comparison of daily bisphenol A intake based on dietary and urinary levels in breastfeeding women. Reprod Toxicol, 2021. 106: 9-17.

[2] Liu, J., et al., Exposure and dietary sources of bisphenol A (BPA) and BPA-alternatives among mothers in the APrON cohort study. Environ Int, 2018. 119: 319-326.

[3] Costa, SA., et al., Methodological approaches for the assessment of bisphenol A exposure. Food Res Int, 2023. 173(Pt 1):113251.

Reviewer:

  1. Line 95: "search for articles focusing on the contamination status, sources and exposure routes of bisphenols published between 2018 and 2023."

Why limit your search to articles published after 2018?

Authors:

We thank for reviewer’s thorough review and insightful questions. Due to the temporal nature of the content in this section, which primarily focuses on the sources, contamination status, and exposure pathways of bisphenol compounds, it is pertinent to note that literature reporting on the contamination status of bisphenols from a decade ago may be considerably disconnected from the current environmental situation. Furthermore, with the introduction of various national policies, the focus of bisphenol A contamination has gradually shifted towards its alternatives. Therefore, conducting a review of research from a significantly earlier period may not be as essential. As a result, we have chosen to primarily select literature from the past five years.

Additionally, during our search process, we encountered challenges in using specific keywords such as "bisphenol status," "bisphenol sources," and "bisphenol exposure" individually, as they were unable to comprehensively cover all relevant literature on these topics. Consequently, we expanded the keyword scope to "bisphenol" to enhance the precision and inclusiveness of our work. However, this expansion also led to the issue of an exceptionally large volume of literature, with over 7300 articles related to the keyword "bisphenol" published in the past five years.

Reviewer:

  1. Line 295: "The urinary bisphenols concentration is highest among cashiers (1.12 ng/mL)"

Question: in previous sentence you list two other professions with higher urinary concentrations?

Authors:

We thank for reviewer’s careful review and constructive questions. We have revised the original statement to address ambiguities in lines 311-313: "Apart from workers directly involved in the manufacturing of epoxy resins and thermal paper, individuals working as cashiers exhibited the highest urinary bisphenol concentrations."

Reviewer:

  1. Line 2: "Pusher" is not correct English.

Suggest change "Invisible Pusher of Female Reproductive Disorders" to "Invisible Determinant of Female Reproductive Disorders"

Authors:

We appreciate the reviewer's suggestion, and we have changed the title to “Invisible Hand behind Female Reproductive Disorders: Bisphenols, Recent Evidence and Future Perspectives”.

Reviewer:

  1. Line 12: Suggest change "The increased exposure to bisphenols has raised..." to

"The increased exposure to bisphenols, a chemical used in large quantities for the production of polycarbonate plastics, has raised..."

Authors:

We are grateful to the reviewers for their careful review and helpful suggestions, and we have revised lines 12 and 13 to your suggested presentation, which makes the expression clearer and more accurate.

Reviewer:

  1. Line 28: Image at bottom of page lacks figure legend. Does not appear to add anything to the understanding of the review. Suggest delete this image.

Authors:

We thank you for reviewer’s careful review and helpful suggestions, and we have added annotations in the figure to enhance our understanding of the graphical abstract.

Reviewer:

  1. Line 42: "endocrine-disrupting chemicals (EDCs) can mimic or block natural hormone activity in the human body."

Citation(s) needed

Authors:

We appreciate reviewer’s careful review and we have added a reference at the end of the sentence on line 43.

Reviewer:

  1. Line 46: "As an EDC, bisphenol A (BPA) has profound adverse effects"

Should briefly mention in introduction what/why BPA exposure is so common (e.g. a chemical used in large quantities for the production of polycarbonate plastics) before outlining its dangers

Authors:

We appreciate your valuable suggestion, this section is really needed in the article and your suggestion is very necessary, we have added in line 46 “Bisphenol A (BPA), a high-yield monomer widely used in the synthesis of poly-carbonate plastics and epoxy resins, has been detected in daily necessities such as canned food and beverages, food container liners, and medical devices, and regulatory agencies across the world have banned the use of BPA due to its serious adverse effects on reproduction, metabolism, development and nervous and immune systems [1]”.

Reference:

[1] Mustieles, V., et al., Bisphenol A and its analogues: A comprehensive review to identify and prioritize effect biomarkers for human biomonitoring. Environ Int, 2020. 144: p. 105811.

Reviewer:

  1. Line 46: "(BPA) has profound adverse effects on reproduction, metabolism, development and nervous and immune systems."

Citation(s) needed

Authors:

We appreciate reviewer’s careful review and we have added the citation at the end of this sentence in line 50.

Reviewer:

  1. Line 63: Suggest change "Gal et al." to "Ziv-Gal et al. "

Authors:

We thank for reviewer’s careful review and correction, and we have changed "Gal et al." to "Ziv-Gal et al." at line 83.

Reviewer:

  1. Lines 68-74: "Administration of BPA/BPS to animals during pregnancy may cause damage to the giant cell neurotransmitter circulation and immune reactivity of placental trophoblasts. Exposure to BPB and BPAF leads to accelerated differentiation of the mammary glands and remarkably increases amino acid levels in the mother. The imbalance of placental and breast functions leads to abnormal foetal development and altered expression of genes related to carcinogenesis in the adult offspring [14-16]."

Please add citations at end of sentences, instead of waiting to group them all at end of paragraph.

Authors:

We thank for reviewer’s careful review, and we have revised the use of citations in the article in lines 89-95.

Reviewer:

  1. Lines 174;219 Tables 1, 2: Reduce line spacing so tables appear on 1 page.

Authors:

We appreciate the reviewer's suggestion, in line 193, we have re-spaced Table 1 so that it can be displayed in its entirety on one page. Since Table 2 has too much information to be placed on one page, it would not be possible to show the complete information, so we only adjusted the spacing and could not place it on one page.

Reviewer:

  1. Line 723 Figure 1: Figure 1 is first mentioned in line 324, but appears at line 723. Figure 1 should be moved so it's closer to when first mentioned in text.

Authors:

We are grateful for the reviewer's suggestion, and we have placed the description of the first occurrence of Figure 1 in line 727.

Reviewer:

  1. Line 335: " in adolescent CD-1 mice exposed to BPB or BPAF for 28 days."

Citation(s) needed

Authors:

We thank for reviewer’s careful review, and we have revised the citations in the article in line 352.

Reviewer:

  1. Line 339: " exposure to BPAF leads to dilation of endometrial glands. "

Citation(s) needed

Authors:

We are grateful for the reviewer's suggestion, and we've changed the quoted portion in line 356.

Reviewer:

  1. Line 348: "during adolescence induced uterine growth, which is a marker of oestrogen exposure. "

Citation(s) needed

Authors:

We appreciate reviewer’s careful review and we have revised the citations in the article in line 364.

Reviewer:

  1. Line 351: "However, exposure to BPAP significantly reduced uterine weight in adolescent CD-1 mice. "

Citation(s) needed

Authors:

We thank for reviewer’s careful review and correction and we have revised the citations in line 366.

Reviewer:

  1. Line 372: "In addition, injection of 40-mg/kg BPA into C57BL6 mice during early pregnancy (0.5– 3.5 days) has been reported to delay implantation and increase the perinatal mortality risk, whereas exposure to 20-mg/kg BPA has been reported to remarkably reduce the number of implantation sites in rats. "

Citation(s) needed

Authors:

We appreciate the reviewer's suggestion, and we have revised the citations in line 393.

Reviewer:

  1. Line 382: "In pregnant rats, orally administered BPA (2.5, 25 and 250 μg/kg/day) results in decreased luminal diameter of uterine arteries and endothelial dysfunction by downregulating the expression of endothelial nitric oxide synthase 3 (NOS3), estrogen receptor α (ERα), and peroxisome proliferator-activated receptor γ (PPARγ), thereby negatively affecting reproductive circulation. "

Citation(s) needed

Authors:

We appreciate reviewer’s careful review and we have revised the citations in the article in line 403.

Reviewer:

  1. Line 416: "Bisphenols as xenoestrogens exert adverse effects on reproductive processes by interfering with receptor binding."

Citation(s) needed

Authors:

We are grateful for the reviewer's suggestion, and we've changed the quoted portion in line 433.

Reviewer:

  1. Line 421: "Perinatal exposure to BPA induces E2-mediated hormone receptor changes in the uterus of older rats, thereby affecting the expression of proteins in- volved in the differentiation of uterine glands. "

Citation(s) needed

Authors:

We thank for reviewer’s careful review and correction and we have revised the citations in lines 437-439.

Reviewer:

  1. Line 447: "another study showed that BPA was preferentially enriched in uterine leiomyoma and adjacent myometrial tissue samples from patients."

Citation(s) needed

Authors:

We appreciate the reviewer's suggestion, and we have revised the citations in line 465.

Reviewer:

  1. Line 461: "rats exposed to TBBPA developed tumoral and non-tumoral lesions in the uterus that resembled high-grade type 1 tumours in women in terms of morphological and molecular characteristics. "

Citation(s) needed

Authors:

We appreciate reviewer’s careful review and we have revised the citations in the article in line 481.

Reviewer:

  1. Line 474: "Jones et al. showed that the differential expression of target genes in EM lesions after exposure to BPA and BPAF (3, 30 and 90 mg/kg/day) was related to the hormone status of mice. "

Citation(s) needed

Authors:

We are grateful for the reviewer's suggestion, and we've changed the quoted portion in line 493.

Reviewer:

  1. Line 481: "can lead to serious uterine lesions, such as cystic endometrial hyperplasia and squamous metaplasia, adenomyosis, leiomyoma, atypical hyperplasia, cervical sarcoma and stromal polyps."

Citation(s) needed

Authors:

We thank for reviewer’s careful review and correction and we have revised the citations in line 500.

Reviewer:

  1. Line 490: Suggest change "epigenetic regulation can reveal the toxic effects of bisphenols" to "epigenetic regulation can be modified by the toxic effects of bisphenols"

Authors:

We appreciate the reviewer's suggestion, and we have We have amended the statement in the article to “epigenetic regulation can be modified by the toxic effects of bisphenols” in lines 507-508.

Reviewer:

  1. Line 493: "DNA methylation, a widely investigated epigenetic mechanism, usually occurs in CpG islands. "

Citation(s) needed

Authors:

We appreciate reviewer’s careful review and we have revised the citations in the article in line 510.

Reviewer:

  1. Line 495: "Continuous exposure to BPA for 1 week during pregnancy significantly induces the expression of the Dnmt3b (a methyltransferase) in uterine stromal cells and increases the mRNA and protein expression of Hoxa1 in uterine tissues in offspring."

Citation(s) needed

Authors:

We are grateful for the reviewer's suggestion, and we've changed the quoted portion in line 513.

Reviewer:

  1. Line 501: "Taylor et al. showed that BPA influenced the DNA methylation pattern of Hoxa10. The effects of BPA on the uterus are intergenerational. "

Citation(s) needed

Authors:

We thank for reviewer’s careful review and correction and we have revised the citations in line 516, and a correction was made to a previous error in the citation of authors’ names.

Reviewer:

  1. Line 521: "H3K4me3 at the promoter region is closely related to the expression of ERβ in the endometrium. "

Citation(s) needed

Authors:

We appreciate the reviewer's suggestion, and we have revised the citations in line 539.

Reviewer:

  1. Line 522: "A recent study showed that exposure of mice with EM to BPA upregulated the expression of WD repeat domain 5 (WDR5) through the G protein-coupled oestrogen receptor (GPER)-mediated PI3K/mTOR signalling pathway, resulting in increased recruitment of the WDR5–tetmethylcytosine dioxygenase 2 (TET2) complex to ERβ. "

Citation(s) needed

Authors:

We appreciate reviewer’s careful review and we have revised the citations in the article in line 543.

Reviewer:

  1. Line 529: "N6 methyladenosine (m6A) plays an important regulatory role in multiple biological processes. "

Citation(s) needed

Authors:

We are grateful for the reviewer's suggestion, and we've changed the quoted portion in line 547.

Reviewer:

  1. Line 532: "Orally administered BPA (0.5, 5 and 50 mg/kg) in 4-week-old SD rats interferes with the expression of METTL3 in the uterus in a dose-dependent manner."

Citation(s) needed

Authors:

We thank for reviewer’s careful review and correction and we have revised the citations in lines 549-552.

Reviewer:

  1. Line 540: "Treatment with BPA (10, 103 and 105 nM) induces the overexpression of miR-107 in the hedgehog signalling pathway and downregulates miR-149 and its related DNA repair genes in the ADP-ribosylation factor 6 (ARF6)-TP53-cyclin E2 (CCNE2) pathway in human endometrial cancer RL95-2 cells. "

Citation(s) needed

Authors:

We appreciate the reviewer's suggestion, and we have revised the citations in line 590.

Reviewer:

  1. Lines 555; 631; 792; 845; 856; 871: Suggest change "In a study" to "In one study"

Authors:

We are grateful for the reviewer's suggestion, and we've changed "In a study" to "In one study" in lines 561, 636, 796, 847, 871 and 976.

Reviewer:

  1. Line 577: "Ko et al. found that low-dose (5 μM) treatment with octylphenol (OP) and nonylphenol (NP) significantly reduced the ability of human endometrial cells to secrete PRL and IGFBP1. "

Citation(s) needed

Authors:

We thank the reviewer for this comment. Octylphenol (OP) and nonylphenol (NP) are non-ionic surfactants used widely as emulsifiers, detergents, solubilisers, antistatic agents, and dispersing agents in domestic, industrial and agricultural products, they are also used in plastic additives and pesticides. We have re-searched the structural formulas of NP and OP, which are outside of the bisphenols we are concerned about, so we have deleted the literature on NP and OP included here.

Reviewer:

  1. Line 579: "Increased expression of the bone morphogenetic protein 2 (BMP2), a key protein involved in decidualization, induced the expression of Wnt4 and controlled the expression of forkhead box O1 (FOXO1) and its downstream gene left-right determination factor 1 (LEFTY1) through the β-catenin pathway. "

Citation(s) needed

Authors:

We thank the reviewer for this comment. Octylphenol (OP) and nonylphenol (NP) are non-ionic surfactants used widely as emulsifiers, detergents, solubilisers, antistatic agents, and dispersing agents in domestic, industrial and agricultural products, they are also used in plastic additives and pesticides. We have re-searched the structural formulas of NP and OP, which are outside of the bisphenols we are concerned about, so we have deleted the literature on NP and OP included here.

Reviewer:

  1. Line 584: "In another study, treatment with 1-μM BPA significantly increased ROS production in human endometrial stromal cells (ESCs)."

Citation(s) needed

Authors:

We appreciate reviewer’s careful review and we have revised the citations in the article in line 582.

Reviewer:

  1. Line 596: "Kang et al. analysed the mRNA expression profiles of leiomyoma cells after 48 and 96 h of BPA exposure. "

Citation(s) needed

Authors:

We are grateful for the reviewer's suggestion, and we've changed the quoted portion in line 604.

Reviewer:

  1. Line 600: "Li et al. showed that exposure to BPA significantly altered the differential expression of 739 genes in uterine leiomyoma cells."

Citation(s) needed

Authors:

We thank for reviewer’s careful review and correction and we have revised the citations in lines 606.

Reviewer:

  1. Line 605: "Wang et al. showed that exposure to BPA enhanced the growth and colony-forming ability of human endometrial cancer cells (RL95-2) in a dose-dependent manner. "

Citation(s) needed

Authors:

We appreciate the reviewer's suggestion, and we have revised the citations in line 611.

Reviewer:

  1. Line 618: "UF is the most common monoclonal tumor of the uterine muscle of the reproductive system in women of reproductive age worldwide. "

Citation(s) needed

Authors:

We appreciate reviewer’s careful review and we have revised the citations in the article in line 624.

Reviewer:

  1. Line 631: "the incidence of UFs and urinary concentrations of BPA and phthalate were measured in 495 women."

Citation(s) needed

Authors:

We are grateful for the reviewer's suggestion, and we've changed the quoted portion in line 637.

Reviewer:

  1. Line 648: "Sampson et al."

Citation(s) needed. No Sampson et al found in reference list.

Authors:

We thank for reviewer’s careful review and correction and we have revised the citations in line 654. The literature we cited mentions "The most widely accepted hypothesis, first advanced by Sampson in 1921, is that viable endometrial tissue fragments move retrograde through the fallopian tubes into the pelvic cavity during menstruation [1]."

Reference

[1] Dutta, S., S.K. Banu, and J.A. Arosh, Endocrine disruptors and endometriosis. Reprod Toxicol, 2023. 115: p. 56-73.

Reviewer:

  1. Line 664: "Peinado et al. evaluated the relationship between EM and the urinary concentrations of BPA, BPS and BPF in 124 women aged 20–54 years."

Citation(s) needed

Authors:

We appreciate the reviewer's suggestion, and we have revised the citations in line 670.

Reviewer:

  1. Line 669: "Cobellis et al. showed that the serum concentrations of BPA and BPB were 2.91 ± 1.74 and 5.15 ± 4.16 ng/mL, respectively, in 58 patients with peritoneal EM of childbearing age. "

Citation(s) needed

Authors:

We appreciate reviewer’s careful review and we have revised the citations in the article in line 674.

Reviewer:

  1. Line 681: "A case–control study conducted by Wen et al. showed that the levels of MMP-2 were 175.98 ng/mL and 145.34 ng/mL and those of MMP-9 were 807.41 ng/mL and 750.74 ng/mL in patients with EM and control individuals, respectively."

Citation(s) needed

Authors:

We are grateful for the reviewer's suggestion, and we've changed the quoted portion in line 687.

Reviewer:

  1. Line 689: "Ehrlich et al. examined the effectiveness of in vitro fertilisation (IVF) in 174 women of childbearing age. "

Citation(s) needed

Authors:

We thank for reviewer’s careful review and correction and we have revised the citations in line 693.

Reviewer:

  1. Line 703: "Wang et al. evaluated the relationship between pre-pregnancy urinary concentration of BPA and fertility in 700 Chinese couples. "

Citation(s) needed

Authors:

We appreciate the reviewer's suggestion, and we have revised the citations in line 708.

Reviewer:

  1. Line 715: "In a case study on recurrent miscarriages (RMs) in human patients, urinary BPA concentrations were positively associated with the risk of RMs."

Citation(s) needed

Authors:

We appreciate reviewer’s careful review and we have revised the citations in the article in lines 719-721.

Reviewer:

  1. Line 986 Figure 2: Figure 2 is first mentioned in line 733, but appears at line 986. Figure 2 should be moved so it's closer to when first mentioned in text.

Authors:

We are grateful for the reviewer's suggestion, and we have adjusted the first reference to Figure 2 in line 980.

Reviewer:

  1. Line 739: "Similar effects, including the presence of atretic follicles, cyst formation, separation of granulosa cells, vascular congestion and increased thickness of the tunica albuginea, have been observed in rats exposed to BPA. "

Citation(s) needed

Authors:

We thank for reviewer’s careful review and correction and we have revised the citations in line 746.

Reviewer:

  1. Line 792: "exposure of pregnant mice to BPA increased the relative expression of the pro-apoptotic genes Caspase-7, Caspase-9 and Bax while decreasing the relative expression of the anti-apoptotic gene Bcl-2, resulting in an increase in the ovarian apoptotic rate in F1 female mice before puberty and adulthood. "

Citation(s) needed

Authors:

We appreciate the reviewer's suggestion, and we have revised the citations in line 800.

Reviewer:

54.Line 814: "The long non-coding RNA Fhad1os2 is aberrantly expressed in the ovaries of adolescent mice exposed to BPA. "

Citation(s) needed

Authors:

We appreciate reviewer’s careful review and we have revised the citations in the article in line 820.

Reviewer:

55.Line 820: "Prenatal exposure to BPA (F0 mice) results in decreased levels of cytochrome P450 side-chain cleavage, 3β-hydroxysteroid dehydrogenase 1 (3βHSD1), and aromatase cytochrome P450 (P450AROM) mRNA in F1 and F2 mice."

Citation(s) needed

Authors:

We are grateful for the reviewer's suggestion, and we have revised the citations in line 825.

Reviewer:

  1. Line 837: "However, exposure to 1-μg/L bisphenols leads to an increase in plasma E2 levels and a decrease in testosterone levels in adult female zebrafish. "

Citation(s) needed

Authors:

We thank for reviewer’s careful review and correction and we have revised the citations in line 841.

Reviewer:

  1. Line 845: "exposure of adult zebrafish to environmentally relevant concentrations of BPA for 30 days promoted the synthesis of reactive oxygen species/reactive nitrogen species (ROS/RNS) and lipid peroxidation in the ovary, thereby reducing the activity of hydrogen peroxide enzyme and exacerbating oxidative stress responses."

Citation(s) needed

Authors:

We appreciate the reviewer's suggestion, and we have revised the citations in line 851.

Reviewer:

  1. Line 856: "oral administration of BPA and BPB in adult laying hens resulted in lymphocyte and plasma cell infiltration in the ovaries. "

Citation(s) needed

Authors:

We appreciate reviewer’s careful review and we have revised the citations in the article in line 860.

Reviewer:

  1. Line 871: "rat ovaries were collected on postnatal day 4 and cultured in a medium containing BPA. "

Citation(s) needed

Authors:

We are grateful for the reviewer's suggestion, and we have revised the citations in the article in line 873.

Reviewer:

  1. Line 874: "in vitro maturation of primordial germ cell–oocyte complexes with or without bisphenols showed that bisphenols decreased the rates of cleavage and blastocyst formation when compared with the control group."

Citation(s) needed

Authors:

We thank for reviewer’s careful review and correction and we have revised the citations in line 878.

Reviewer:

  1. Line 959: "POR typically refers to diminished ovarian reserve or poor response of the ovaries to exogenous gonadotropin (Gn) stimulation. It is characterised by a low number of developing follicles during ovarian stimulation cycles, low peak oestradiol levels, high Gn dosage requirements, high cycle cancellation rates, low oocyte yields and low clinical pregnancy rates."

Citation(s) needed

Authors:

We appreciate the reviewer's suggestion, and we have added the citations in line 960.

Reference

[1] Cedars MI. Managing poor ovarian response in the patient with diminished ovarian reserve. Fertil Steril. 2022 Apr;117(4):655-656. doi: 10.1016/j.fertnstert.2022.02.026. PMID: 35367010.

Reviewer:

  1. Line 979: "In a study involving 110 girls with precocious puberty and 100 healthy girls, BPA was detected in 40.9% of blood samples from girls with precocious puberty and 5% of blood samples from healthy girls."

Citation(s) needed

Authors:

We are grateful for the reviewer's suggestion, and we have revised the citations in line 979.

Round 2

Reviewer 2 Report

Comments and Suggestions for Authors

All my comments were included in the text

Author Response

Journal: Toxics

Manuscript ID: toxics-2732595

Title: Invisible Hand behind Female Reproductive Disorders: Bisphenols, Recent Evidence and Future Perspectives

Dear reviewer,

Thanks for reviewing our manuscript “Invisible Hand behind Female Reproductive Disorders: Bisphenols, Recent Evidence and Future Perspectives”. Your comments are all valuable and very helpful for revising and improving our paper, as well as the important guiding significance to our researches. On behalf of my co-authors, we would like to express our great appreciation to you.

Kind regards,

Huifeng Yue, Ph.D.

College of Environment and Resource, Shanxi University

Taiyuan, Shanxi 030006, People’s Republic of China

Reviewer 3 Report

Comments and Suggestions for Authors

Authors have adequately addressed reviewer concerns.

Author Response

(The authors gave the same response as above.)
